# Enhanced Detection Performance of Acute Vertebral Compression Fractures Using a Hybrid Deep Learning and Traditional Quantitative Measurement Approach: Beyond the Limitations of Genant Classification

**DOI:** 10.3390/bioengineering12010064

**Published:** 2025-01-13

**Authors:** Jemyoung Lee, Minbeom Kim, Heejun Park, Zepa Yang, Ok Hee Woo, Woo Young Kang, Jong Hyo Kim

**Affiliations:** 1Department of Applied Bioengineering, Graduate School of Convergence Science and Technology, Seoul National University, Seoul 08826, Republic of Korea; jaymlee0407@snu.ac.kr; 2ClariPi Research, ClariPi Inc., Seoul 03088, Republic of Korea; mbkim3014@claripi.com; 3Department of Radiology, Korea University Guro Hospital, Seoul 08308, Republic of Korea; eirbadmin@kumc.or.kr (H.P.); yangzepa@gmail.com (Z.Y.); wokhee@korea.ac.kr (O.H.W.); 4Department of Radiology, Seoul National University College of Medicine, Seoul 03080, Republic of Korea; 5Department of Radiology, Seoul National University Hospital, Seoul 03080, Republic of Korea; 6Center for Medical-IT Convergence Technology Research, Advanced Institutes of Convergence Technology, Suwon 16229, Republic of Korea

**Keywords:** acute vertebral compression fracture, genant classification, deep learning, computed tomography, spine

## Abstract

Objective: This study evaluated the applicability of the classical method, height loss ratio (HLR), for identifying major acute compression fractures in clinical practice and compared its performance with deep learning (DL)-based VCF detection methods. Additionally, it examined whether combining the HLR with DL approaches could enhance performance, exploring the potential integration of classical and DL methodologies. Methods: End-to-End VCF Detection (EEVD), Two-Stage VCF Detection with Segmentation and Detection (TSVD_SD), and Two-Stage VCF Detection with Detection and Classification (TSVD_DC). The models were evaluated on a dataset of 589 patients, focusing on sensitivity, specificity, accuracy, and precision. Results: TSVD_SD outperformed all other methods, achieving the highest sensitivity (84.46%) and accuracy (95.05%), making it particularly effective for identifying true positives. The complementary use of DL methods with HLR further improved detection performance. For instance, combining HLR-negative cases with TSVD_SD increased sensitivity to 87.84%, reducing missed fractures, while combining HLR-positive cases with EEVD achieved the highest specificity (99.77%), minimizing false positives. Conclusion: These findings demonstrated that DL-based approaches, particularly TSVD_SD, provided robust alternatives or complements to traditional methods, significantly enhancing diagnostic accuracy for acute VCFs in clinical practice.

## 1. Introduction

### 1.1. Vertebral Compression Fracture (VCF)

VCF is a common outcome of declining bone density, particularly in elderly patients, with postmenopausal individuals being especially vulnerable due to significant bone loss [1]. According to the “Osteoporosis and Osteopenia Fractures Fact Sheet 2023” published by the Korean Society for Bone and Mineral Research, the incidence of vertebral fractures increased sharply with age, with more than 50.5% of patients aged 70 and above experiencing vertebral fractures. Furthermore, as of 2021, the overall re-fracture rate at any location within one year after an initial vertebral fracture rose to 8.7%. In the United States, an estimated 1.5 million occurrences of VCFs are documented annually [2]. With an increasingly aging population, the incidence of VCFs was expected to continue rising steadily. VCFs were the most frequent type of osteoporotic fracture and were associated with significant morbidity and mortality. Early diagnosis was crucial to prevent complications; however, VCFs were often challenging to identify, as they could be asymptomatic or present with vague symptoms [3,4,5].

### 1.2. Classification of VCF

The Genant classification approach, a traditional quantitative method for measuring VCFs, was employed in most research studies [6]. Three different locations were used to assess each vertebral height: anterior, middle, and posterior. The posterior height of each vertebra was compared with its anterior and middle heights. The height loss ratio (HLR) of each vertebra was determined by the following equation:$$HLR=\left(1-\frac{\min\left({Height}_{anterior},\;{\;Height}_{middle}\right)}{{Height}_{posterior}}\right)*100\;[\%]$$

However, the Genant classification necessitated the placement of three vertical reference lines on each vertebra, making it a time-intensive process. Identifying fractures within bone imaging required meticulous manual analysis, often carried out by experienced radiologists [7]. Additionally, not all vertebrae were clinically categorized using the Genant classification. Although the HLR-based Genant classification provided a framework for diagnosing VCFs, it was insufficient as a definitive criterion. In clinical practice, distinctions between acute and chronic VCFs were more commonly employed for diagnostic purposes [8].

In the acute VCF (typically within the first 6–8 weeks after occurrence), fracture lines were relatively well-defined, and the recent collapse of the vertebral endplates and cortical margins resulted in sharp delineation. At the fracture site, the cortical bone appeared abruptly interrupted or angular, with no evidence of bone bridging [9,10]. The vertebral body often displayed a distinct “step-off” or sharp angulation at the site of the fracture. Due to the absence of bony unions, the collapsed regions lacked signs of new bone formation or sclerosis [11,12]. Deformity progression remained possible, and subtle changes in vertebral alignment occurred with postural adjustments or when mechanical stability was not fully restored.

In chronic VCF, the deformity stabilized and became long-standing. Over time (usually several months), the fracture line gradually became less distinct. The remodeling process, driven by osteoclastic resorption followed by osteoblastic activity, blurred, smoothed, or even completely obliterated the initial fracture line [13,14]. Mature fractures often appeared as stable and well-integrated deformities without a clearly visible fracture line. New bone formation along the endplates and vertebral margins led to sclerotic changes, giving the vertebral body a more uniform and remodeled appearance [13]. The vertebra displayed thickened cortical edges and more regular, rounded contours at the site of the previous fracture. This remodeling reflected a healing response, indicating that the fracture was no longer active.

Vertebral fractures were categorized into acute and chronic types. The vertebral height loss calculation method offered the advantage of quantitatively evaluating vertical morphological deformation. However, this approach appeared to have limitations in effectively detecting clinically significant acute fractures, which exhibit more complex deformities and morphological features, in real-world clinical settings. Furthermore, the review of previous studies revealed that research considering acute fractures in the detection of vertebral compression fractures has been underestimated and is relatively rare.

### 1.3. Related Works

Numerous studies have been conducted to develop DL models based on radiographs; plain radiographs were typically used as the first-line investigation for diagnosing VCFs. However, acute VCFs were not specifically considered in these studies.

Cheng L. et al. detected each vertebra on preprocessed lateral spine X-ray images and segmented each vertebral body solely for the classification of normal vertebrae, compression fractures, and burst fractures [15]. This study utilized the YOLOv4 and ResUNet models, achieving a precision of 74% in identifying compression fractures; however, the acute state of the fractures was not considered. Seo J. W. et al. developed an automated method for measuring vertebral body compression exclusively in the vertical direction from X-ray images using segmentation and regression based on a convolutional neural network (CNN) [16]. This study focused solely on the vertical compression of the vertebral body. Murata K. et al. introduced a deep convolutional neural network (DCNN) using the Visual Recognition V3 model to classify VCF patients from plain thoracolumbar radiography [17]. Their model achieved 84.7% sensitivity and 87.3% specificity in a dataset of 300 patients; however, the acute state of the fractures was not specified. Guermazi A. et al. assessed the impact of the DL model, based on Detectron2, on the diagnostic performance of physicians [18]. Although this study exclusively considered acute fractures, the dataset of 480 radiographs included various body parts beyond the spine, such as the ankle, knee, leg, hip, wrist, and elbow. Chen H. Y. et al. employed a ResNet-based DL model to evaluate the feasibility of detecting and localizing VCFs from plain abdominal frontal radiographs. Also, this study was limited to the Genant classification method [19].

However, the two-dimensional lateral projection of radiographs often failed to accurately depict the detailed features of VCFs in individual vertebrae. Factors such as the overlapping of ribs in the thoracic and thoracolumbar junction regions, as well as interference from internal organs like the liver and intestines, hindered accurate diagnosis. Therefore, CT or MRI was generally considered the reference standard for diagnosing VCFs. Although MRI was capable of detecting edema and soft tissue lesions related to VCFs, its longer acquisition time often resulted in a preference for CT. VCFs could be effectively identified in sagittal-reformatted CT images, where cortical bone deformation and variations in trabecular bone Hounsfield Units (HU) were clearly visible.

Baum T. et al. investigated the feasibility of using the anterior-posterior ratio and middle-posterior ratio for diagnosing osteoporotic vertebral fractures, utilizing only the Genant classification method [20]. Tomita N. et al. employed CNN and long short-term memory (LSTM) networks to detect VCFs on CT examinations [21]. All 129 CT scans in the test set were established as reference standards using the Genant classification. Li Y. et al. focused on evaluating benign and malignant vertebral fractures using a ResNet architecture [22]. Osteoporotic compression fractures were categorized as benign fractures, whereas malignant fractures were associated with metastatic cancer. However, the acute state of the fractures was not identified. Also, acute VCFs were not specifically considered in previous studies with CT images.

Clinically, detecting acute VCFs is crucial [8]. However, most previous studies focused solely on classifying VCFs based on the Genant grade, which relies on height compression ratios, and exploring methodologies for this classification. It remains necessary to verify whether the existing height loss measurement methods are effective for detecting acute compression fractures that are clinically significant. Developing new approaches for detecting acute compression fractures represents an unmet need in the current research landscape.

### 1.4. Purpose and Proposed Approaches

This study evaluated the applicability of the HLR, traditionally limited to assessing vertical spinal height deformation, for detecting major acute compression fractures in clinical settings, while identifying its limitations. To enhance the detection performance of acute VCFs, a novel DL-based method was proposed that utilized the full structural information of the vertebral body, rather than considering only vertical deformations. This approach aimed to demonstrate the potential of DL in detecting acute VCFs. As part of this investigation, a DL model based on vertebral contour information—a concept not previously explored—was introduced to examine its utility in detecting fractures, including horizontal deformations. This model highlighted the applicability of DL in addressing the limitations of traditional quantitative HLR methods. Finally, an integrated DL detection process was proposed, combining HLR-based measurements with the newly developed DL model to overcome the limitations of existing approaches.

To achieve these objectives, the commercial software, ClariVBA (Ver1.0, ClariPi Inc., Seoul, Republic of Korea), for calculating vertebral HLR and newly developed DL techniques was implemented. The respective performances of these methods in detecting acute VCFs were then evaluated. Additionally, the detection performance of the combined approach, integrating HLR measurements with DL methods, was compared and assessed. For the DL-based detection of acute VCFs, three distinct methodologies were applied. These included an end-to-end model for directly detecting fractures from input images, a two-stage method that first segmented vertebral body contours and subsequently detected fractures within the segmented regions, and another two-stage model that identified individual vertebral bodies and classified them as either having VCFs or not. A comparative analysis of these methodologies was conducted to identify the most effective DL approach for this purpose. This structured investigation emphasized the potential of integrating traditional quantitative measurements with advanced DL techniques to improve the detection and evaluation of acute VCFs.

## 2. Materials and Methods

### 2.1. Proposed Methods

This study aimed to compare the current core method for diagnosing VCFs, which relied on the measurement of the HLR, and to investigate performance changes in acute VCF diagnosis through the application of various DL techniques. First, for the traditional height loss measurement based on the Genant classification, we utilized the commercial software ClariVBA (Ver1.0, ClariPi Inc., Seoul, Republic of Korea), which automatically measured the anterior, middle, and posterior heights of each vertebra.

The Genant classification method was utilized to assess the height of three specific regions in each vertebral body. According to this system, vertebrae were classified into four categories: normal (HLR < 20%), mild compression (20% ≤ HLR < 25%), moderate compression (25% ≤ HLR < 40%), and severe compression (HLR ≥ 40%) [6]. The ClariVBA program automatically measured the height of the three regions for each vertebral body, calculated the height loss ratio, and provided quantitative values adjacent to each vertebra. Moderate fractures were displayed in orange, severe fractures in red, and mild fractures, which could exhibit inter- or intra-observer variation, were displayed in the same green color as normal vertebrae (Figure 1). In this study, vertebrae with mild compression were not classified as VCFs.

The DL-based methods were divided into an End-to-End approach and a Two-Stage approach, with the latter further categorized into Two-Stage Type SD (SD stands for Segmentation and Detection) and Two-Stage Type DC (DC stands for Detection and Classification), depending on the specific deep learning methodology employed.

The End-to-End approach employed the YOLO model, which was trained to generate bounding boxes for regions with VCF using preprocessed CT images as input (Figure 2). Hereafter, we referred to this method as EEVD; End-to-End VCF Detection. A model designed to directly detect VCFs from input images using DL was regarded as the simplest approach to VCF detection. This method trained a DL model to identify VCFs directly from input images without requiring complex processing steps, mimicking the way a human expert would analyze the images. By focusing solely on determining the location of VCFs in the input images, the utility of the trained DL model was evaluated.

The Two-Stage Type SD method followed the process described in our previous study and involved two main steps [23]. First, segmenting only the spinal region through spine segmentation, and then extracting the contour of the segmented spine, followed by training a detection model to generate bounding boxes for regions with VCFs (Figure 3). We referred to this method as TSVD_SD, i.e., Two-Stage VCF Detection Type Segmentation and Detection. This method focused on detecting VCFs by emphasizing vertebral contour information, which is a key feature of acute VCFs, through vertebral segmentation. The model was designed to restrict other information outside the vertebral contour, enabling detection based on the irregularities in the vertebral contour regions. Although features from the surrounding vertebral areas or internal vertebral structures could potentially aid the model’s training process, the morphological deformation of the vertebral contour, as observed in CT images, was identified as the critical characteristic of acute VCFs, leading to the development of this model.

In the Two-Stage Type DC approach, the initial step involved vertebra detection, in which the YOLO model was used to identify individual vertebrae (Figure 4). Subsequently, the vertebrae located within the detected bounding boxes underwent classification to determine whether they were acute VCFs or not, thereby identifying the VCF. We referred to this method as TSVD_DC, i.e., Two-Stage VCF Detection Type Detection and Classification. By pre-identifying the location of each vertebra using bounding boxes, it was possible to evaluate the presence of VCFs while considering only the individual vertebral regions. This approach enabled the development of a classification model that could learn not only the morphological features of vertebrae but also the characteristics of the trabecular bone region within the vertebrae by training on cropped images of individual vertebrae.

All of our proposed deep learning models were developed using PyTorch 2.1.0 on a GTX 3090 GPU (Nvidia, Santa Clara, CA, USA).

### 2.2. Patient Datasets

This retrospective study was authorized by the Institutional Review Board (IRB) of Korea University Guro Hospital and conformed to the Helsinki Declaration. The study cohort for benign VCFs was created by searching the institutional Picture Archiving and Communication System (PACS) database for CT scans of the lumbar or thoracic spine performed between August 2018 and July 2021. Reports containing the keywords “compression fracture”, “burst fracture”, or “compression deformity” were included in the search. A total of 1045 patients were initially identified. The exclusion criteria were as follows: (1) any postoperative or procedural state involving spinal surgery, including the insertion of bone cement or metallic hardware; (2) pathological compression fractures caused by infections or tumors; (3) cervical spine or sacral fractures; (4) duplicate imaging of the same patient; and (5) external imaging.

After reviewing the cases, two seasoned radiologists with a combined 20 years of experience in musculoskeletal imaging, along with a third-year radiology trainee, categorized benign VCFs into acute and chronic types. Osteoporotic or traumatic fractures unrelated to cancer were referred to as benign VCFs. Symptoms such as cortical breakage, step-off, disruption, low-attenuation fissures, or sclerotic lines on CT were used to identify acute VCFs.

This study focused solely on acute VCFs, with a total of 589 patients meeting the inclusion criteria (Figure 5). Both normal and acute VCFs were thoroughly analyzed at the level of the vertebral body. A total of 1030 vertebrae were employed for performance evaluation: the test dataset included 882 normal vertebrae and 148 acute fractures, while the training dataset included 536 instances of acute fractures from 435 patients. This training dataset was used to train the EEVD model, the detection model in TSVD_SD, and the classification model in TSVD_DC.

By constructing bounding box masks around the VCFs, the radiologists identified all VCFs on sagittal-reformatted CT scans. CT images were acquired using five different CT manufacturer models: SOMATOM Force, SOMATOM Definition AS+, and SOMATOM Definition Edge (Siemens Healthineers, Erlangen, Germany); Brilliance 64 (Philips Healthcare, Best, The Netherlands); and Aquilion ONE (Canon Medical Systems, Otawara, Japan). The scans utilized nine different reconstruction kernel types with a 2 mm slice thickness and no interval gap. All selected images were saved in the PACS using bone settings (window width of 1500 and window level of 300).

### 2.3. Deep Learning Model Development

#### 2.3.1. EEVD (End-to-End VCF Detection)

For end-to-end vertebral compression fracture (VCF) detection, we employed the YOLOv9 model [24]. One of the primary advantages of the YOLO model was its exceptional speed, which resulted from processing the entire image in a single pass for object detection. Unlike traditional models that examined distinct regions of an image repeatedly, YOLO scanned the entire image at once, making simultaneous predictions about the positions and categories of objects. Additionally, YOLO typically demonstrated a lower false positive rate compared to other detection models, as it minimized the likelihood of inaccurate detections by considering the entire image when identifying object placements.

The VCF detection model was trained using data from 435 patients, who were randomly divided into a 3:1 ratio from the total of 589 VCF patients. Since only the slices displaying the spine were necessary for training, we first excluded the front and back slices that represented the external body regions from the 139,635 sagittal CT scan slices of these 435 patients. As a result, approximately 25% of the slices were removed, leaving 106,443 slices. Using data augmentation techniques, including translation and random rotation of up to 10 degrees, we ultimately obtained 319,329 slices for training. We applied windowing with a window level of 300 and a width of 850. The YOLOv9c model was trained with a batch size of 4 for 100 epochs, using the stochastic gradient descent (SGD) optimizer. The validation performance of the training dataset, expressed as mean average precision (mAP_0.5:0.95), was 0.991.

#### 2.3.2. TSVD_SD (Two-Stage VCF Detection Type Segmentation and Detection)

In this study, we applied the two-stage VCF detection algorithm, originally developed for chronic fractures in our previous research, to the detection of acute VCFs [23]. Using the U-Net Transformer (UNETR) spine segmentation model, we created a spine mask for each vertebra. Following spine segmentation, we extracted the contour mask of the vertebral body and trained the YOLOv9 VCF detection model.

The spine segmentation model was trained on 324 cases (31,971 slices) from the VerSe2020 dataset, which included vertebral masks for each spinal level and featured CT scans from multiple manufacturers and models [25,26,27]. The VerSe2020 dataset was divided into training (243 cases) and testing (81 cases) using a 3:1 ratio.

For the VCF detection model, we used CT datasets from 435 patients for training, with spine contour masks generated by the segmentation model, and reserved 154 cases for testing. The combination of these models enabled accurate segmentation and vertebral fracture detection, effectively handling diverse anatomical variations in cortical bone deformities.

#### 2.3.3. TSVD_DC (Two-Stage VCF Detection Type Detection and Classification)

We first detected each vertebral region using a bounding box. In the sagittal view of the spine, multiple vertebral bodies were visible, and we designated each region of interest with bounding boxes. We established the vertebra detection training dataset. Since the vertebra detection model was designed to locate vertebral regions, it was possible to generate the training data without reference standards from radiologists. To create the training dataset, a bounding box containing the vertebral body was generated based on the end plate cortical edge from sagittal patient images.

Out of 589 patient datasets from Korea University Guro Hospital, 200 patients were randomly sampled, and dataset creation was conducted on 2640 slices where vertebral regions were identified. The training, validation, and testing splits were performed at a ratio of 8:1:1, resulting in 2135, 253, and 252 slices, respectively, with no additional data augmentation. For image preprocessing, the input images were adjusted using a window level of 300 and a window width of 850. We trained the YOLOv9 architecture with a batch size of 4 for 100 epochs, using the SGD optimizer. The validation performance of the training dataset, expressed as mean average precision (mAP_0.5:0.95), was 0.943.

The vertebral regions identified by the vertebra detection model were cropped, and preprocessing was conducted on these cropped regions. Using the Pillow (version 11.0.0) Python library, the contrast of the input images was augmented by a factor of two to enhance edge features for learning. The model structure used for VCF classification training was ResNet-50, with pre-trained weights from ImageNet. The reason for using ResNet-50 was that the number of parameters in the YOLOv9c structure used for comparison was approximately 25.5 million, and the similar ResNet-50 structure had 25,557,032 parameters. To appropriately utilize the pre-trained weights, the cropped images were scaled to a size of (224, 224) using cubic interpolation, ensuring consistency in input image size. This resizing was performed using the OpenCV (version 1.18.0) Python library.

We first extracted only the vertebral regions from the 435 patients in the training dataset classified from Korea University Guro Hospital using vertebra detection. Cropped vertebra images were labeled as 1 if a reference standard marked by a radiologist was present, and labeled as 0 otherwise, to facilitate training for acute VCF classification. Only the middle slices of the CT images were used, resulting in a total of 55,630 cropped vertebra images. The dataset was randomly divided into training and validation sets at a ratio of 9:1. Data augmentation was limited to random rotation of up to 10 degrees. We trained the ResNet-50 architecture with a batch size of 16 for 100 epochs, using the SGD optimizer.

### 2.4. Statistical Analysis

True positive (TP), true negative (TN), false positive (FP), and false negative (FN) values were calculated using the radiologist reference standard. The vertebra-wise categorization results for acute VCF detection served as the basis for determining the TP, TN, FP, and FN values. For VCF detection performance at each vertebra, we also evaluated sensitivity, specificity, accuracy, and precision, along with the 95% confidence interval (95% CI), using the confusion matrix. Statistical analysis was conducted using the Python libraries scikit-learn (version 0.23.2) and statsmodels (version 0.14.1).

The performance of acute VCF detection was assessed graphically using the receiver operating characteristic (ROC) curve. Across all potential threshold values, this curve compared the true positive rate (sensitivity) to the false positive rate (1-specificity). The area under the curve (AUC), with values ranging from 0 to 1, measured the model’s overall ability to discriminate between positive and negative classes. While the AUC provided a single scalar value that indicated the model’s overall prediction accuracy, the ROC curve allowed for a visual evaluation of the model’s ability to discriminate between the two classes, thereby aiding efficient threshold setting and model comparison.

In this study, we conducted an evaluation to compare the effectiveness of the traditional HLR methodology (Genant Classification) in detecting acute VCFs against DL-based methods, and to determine how much the detection performance of acute VCFs could be improved by complementing the HLR results with DL methods. First, in evaluation method 1, we evaluated the performance of each method—HLR, EEVD, TSVD_SD, and TSVD_DC—individually, without combining them, to determine how effectively each approach identified acute VCFs. Next, in evaluation method 2, we assessed how the performance changed when a DL-based method was used complementarily to adjust cases that were initially judged as negative by HLR; HLR(−). Specifically, we investigated whether DL could correctly identify as TP those cases that HLR misclassified as FN. Finally, in evaluation method 3, we used a DL-based method complementarily to adjust cases judged as positive by HLR; HLR(+) was used to verify whether DL could identify as TN in the cases that HLR misclassified as FP.

## 3. Results

The HLR confusion matrix revealed a relatively low TP (102) compared to the TN (800) (Figure 6). The FP (82) was moderately high, potentially leading to overdiagnosis. In the EEVD, the TP (100) slightly decreased, but this was accompanied by a significant reduction in FP (7). The TN (875) was also relatively high, effectively avoiding false alarms. However, the FN (48) remained high, indicating that the sensitivity needed improvement to reduce missed true positive cases. The TSVD_SD confusion matrix revealed an improved balance with a significant increase in TP (125) and a reduction in FN (23). However, the FP (28) was higher compared to the EEVD model, suggesting an increased likelihood of false positives. The TSVD_DC also demonstrated balanced performance, with a TP (115) and a TN (858). The FN (33) was moderate, suggesting that the model did miss some positive cases but performed reasonably well in avoiding underdiagnosis. The FP (24) was also moderate.

The TSVD_SD demonstrated the highest sensitivity at 84.46% (95% CI: 77.76–89.42%), indicating a strong capacity to detect true positives (Table 1). In contrast, HLR and EEVD showed lower sensitivities of 68.92% (95% CI: 61.06–75.82%) and 67.57% (95% CI: 59.66–74.58%), respectively. The TSVD_DC approach also performed well, achieving a sensitivity of 77.70% (95% CI: 70.34–83.66%). The EEVD achieved the highest specificity at 99.21% (95% CI: 98.37–99.62%), indicating exceptional performance in avoiding false positives. TSVD_SD and TSVD_DC also exhibited high specificity values of 96.83% (95% CI: 95.45–97.79%) and 97.28% (95% CI: 95.98–98.16%), respectively, reflecting robust discrimination between true negatives and false positives. The HLR method, although slightly lower, maintained an acceptable level of specificity at 90.70% (95% CI: 88.61–92.45%). The TSVD_SD approach yielded the highest accuracy at 95.05% (95% CI: 93.55–96.21%), followed closely by EEVD and TSVD_DC at 94.66% (95% CI: 93.11–95.87%) and 94.47% (95% CI: 92.90–95.70%), respectively. HLR demonstrated a lower accuracy at 87.57% (95% CI: 85.42–89.45%). The EEVD also achieved the highest precision at 93.46% (95% CI: 87.11–96.80%), indicating a high level of reliability in identifying positive cases, with fewer false positives. TSVD_ SD and TSVD_ DC performed well in this regard, with precision values of 81.70% (95% CI: 74.82–87.02%) and 82.73% (95% CI: 75.59–88.11%), respectively. However, HLR demonstrated the lowest precision at 55.43% (95% CI: 48.21–62.43%), suggesting a higher likelihood of false positives compared to the other methods.

In evaluation method 2, the TP count of HLR(−) + EEVD was 120, indicating a strong ability to identify positive cases (Figure 7). However, the FP (87) was relatively high, indicating a tendency for overdiagnosis. The TN (795) was also high, suggesting that the model exhibited reasonable specificity in correctly identifying negative cases. The HLR(−) + TSVD_SD demonstrated an increase in TP (130), reflecting improved sensitivity compared to the HLR(−) + EEVD model. This version also had a reduced FN (18). However, the FP increased to 98, suggesting a potential drawback in terms of specificity. The TN (784) was lower compared to the HLR(−) + EEVD model, implying that while the model improved its ability to detect positive cases, this improvement came at the cost of reduced specificity. The HLR(−) + TSVD_DC model exhibited a TP count of 126 and an FN count of 22. The FP (94) and TN (788) suggested that the model attempted to achieve a middle ground, reducing false positives compared to the HLR(−) + TSVD_SD. HLR(−) + TSVD_DC appeared to offer a balance between sensitivity and specificity, resulting in a robust classification performance that minimized both false negatives and false positives to an extent.

The HLR(−) + TSVD_SD achieved the highest sensitivity at 87.84%, followed by the HLR(−) + TSVD_DC at 85.14% (Table 2). The HLR(−) + EEVD, while maintaining a respectable sensitivity of 81.08%, was slightly less effective compared to the TSVD methods, indicating a greater likelihood of missing positive cases. The HLR(−) + EEVD performed slightly better in terms of specificity, with a value of 90.14%, suggesting a reduced risk of generating false positives compared to the TSVD-based methods. The HLR(−) + TSVD_SD and HLR(−) + TSVD_DC demonstrated comparable specificity values of 88.89% and 89.34%, respectively, indicating their effectiveness in avoiding the over-classification of negative cases as positive. The HLR(−) + EEVD achieved the highest accuracy at 88.83%. In terms of overall classification performance, all three models performed similarly, with minor differences that were not statistically significant given the overlapping CIs. The HLR(−) + EEVD also achieved the highest precision at 57.97%, suggesting that it produced fewer false positive results compared to the other two methods. However, the differences in precision among the three models were minor. The HLR(−) + TSVD_SD and HLR(−) + TSVD_DC had precision values of 57.02% and 57.27%, respectively, indicating a similar level of reliability in positive classification. The overlapping CIs further indicated that the models did not show a statistically significant difference in precision.

In evaluation method 3, the HLR(+) + EEVD demonstrated a very low FP (2), indicating exceptional precision in avoiding false positives (Figure 8). The model’s specificity was extremely high, and the sensitivity was relatively low, as evidenced by a high FN (66), which indicated that the model failed to detect a significant number of true positive cases. The low TP (82) further highlighted this limitation, making the model less ideal for applications requiring high sensitivity. The HLR(+) + TSVD_SD showed an improvement in sensitivity compared to the EEVD, with an increased TP (97) and a decreased FN (51). However, the FP also increased to 12, suggesting a trade-off between achieving better sensitivity and reducing false positives. The TN count remained high (870), indicating good specificity. The HLR(+) + TSVD_DC was similar to HLR(+) + TSVD_SD in terms of FP (12) and TN (870) counts, suggesting that specificity remained largely the same between these two versions. However, there was a slight reduction in TP count (91) and an increase in FN count (57) compared to the HLR(+) + TSVD_SD model, indicating a decrease in sensitivity.

The HLR(+) + TSVD_SD exhibited the highest sensitivity at 65.54%, followed by the HLR(+) + TSVD_DC model at 61.49% (Table 3). The HLR(+) + EEVD demonstrated the lowest sensitivity at 55.41%, indicating a greater likelihood of missing true positive cases. The HLR(+) + EEVD model achieved the highest specificity at 99.77%, suggesting a very low rate of false positives, which could minimize unnecessary follow-ups. Both HLR(+) + TSVD_SD and HLR(+) + TSVD_DC had specificity values of 98.64%, which were also high but slightly lower compared to the HLR(+) + EEVD. The HLR(+) + TSVD_SD achieved the highest accuracy at 93.88%, suggesting a balanced approach to classifying both true positive and true negative cases. The HLR(+) + EEVD and HLR(+) + TSVD_DC both demonstrated similar levels of accuracy, at 93.40% and 93.30%, respectively. This indicated that while HLR(+) + TSVD_SD slightly outperformed the others in terms of overall accuracy, the differences among the three models were minimal, as evidenced by overlapping confidence intervals. The HLR(+) + EEVD stood out with the highest precision at 97.62%, suggesting that when it predicted a positive case, it was correct nearly all of the time. The HLR(+) + TSVD_SD and HLR(+) + TSVD_DC models had lower precision values of 88.99% and 88.35%, respectively, indicating a higher rate of false positives compared to the HLR(+) + EEVD.

The TSVD_SD, with an area under the ROC curve (AUROC) of 0.90, was clearly the best standalone model for achieving high sensitivity and specificity, indicating a strong ability to effectively discriminate between positive and negative cases (Figure 9). The negative combinations, specifically HLR(−) combined with TSVD_SD and HLR(−) combined with TSVD_DC, both achieved an AUROC of 0.87, outperforming the standalone HLR model. This outcome suggested that incorporating features from TSVD significantly enhanced the model’s discriminative power. These models were particularly suitable for applications requiring a high level of accuracy, such as medical diagnostics, where minimizing both false positives and false negatives was crucial. The positive combinations, particularly those involving the EEVD (AUROC of 0.77), did not perform as well, in some cases yielding worse performance compared to the HLR alone. This finding indicated that the addition of certain features may have introduced redundancy or conflict within the feature space, thereby reducing overall performance. It highlighted the need for careful consideration when combining models, as not all configurations would necessarily lead to performance improvements. The EEVD, whether used alone or in combination, consistently performed better than the baseline HLR model but lagged behind the TSVD-based models. This suggested that, while EEVD could enhance performance, its contributions were not as impactful as those provided by the TSVD-based enhancements.

Figure 10, Figure 11, Figure 12, Figure 13 and Figure 14 illustrate typical examples of results for HLR, EEVD, TSVD_SD, and TSVD_DC in patients with acute VCFs. The results of applying different methodologies to the same patient image and identical slice locations were as follows. In the HLR results, each vertebra was marked with yellow solid lines indicating the height measurement lines for the anterior, middle, and posterior regions. The measured height values and HLR percentages were displayed on the right side of the image. If the HLR percentage ranged between 25% and 40%, it was highlighted in orange, while values of 40% or higher were highlighted in red. In the EEVD results, the VCF detection outputs were displayed along with their respective confidence scores based on the input patient images. For the TSVD_SD results, after performing spine segmentation, the model extracted only the spine contour, and the VCF detection results with their corresponding confidence scores were presented. In the TSVD_DC results, bounding boxes were generated for each vertebral body, with normal vertebrae represented in green and vertebrae identified as VCFs displayed in red.

In Figure 10, the acute VCF was present only at the L2 level, characterized by morphological features of an anterior cortical step-off. With the HLR method, the result was an FN, whereas EEVD, TSVD_SD, and TSVD_DC correctly identified it as a TP.

In Figure 11, the acute VCF was also present only at the L2 level, with minimal height loss but evidence of cortical breakage at the anterior upper region. In the HLR method, the result was an FN, while EEVD, TSVD_SD, and TSVD_DC correctly detected it as a TP. In addition, HLR produced FP at the T11 and T12 levels. The T11 vertebra was potentially misinterpreted as a VCF due to its wedge-like deformation; however, the radiological assessment concluded it was normal. Both T11 and T12 were classified as VCFs by HLR based on height measurements exceeding 25% and were classified as moderate VCFs according to the Genant classification.

In Figure 12, the L1 vertebra was identified as having an acute VCF, while L4 was determined to be normal. At L1, there was cortical breakage and step-off at the anterior lower region. The HLR method failed to detect this VCF, while EEVD, TSVD_SD, and TSVD_DC successfully identified it. In the HLR assessment, L4 exhibited physiological wedging, with the posterior height measuring more than 25% lower than the anterior and middle heights. Due to the S-shaped curvature of the human spine, it is common for the posterior height to be slightly lower in the L4–L5 region. Consequently, EEVD, TSVD_SD, and TSVD_DC did not classify L4 as a VCF (TN).

In Figure 13, the L1 vertebra was normal, whereas a VCF was present at the L2 level. This patient exhibited irregular cortical endplate appearances from T11 to L2. Radiological evaluation concluded that there was an acute VCF only at the L2 level, which the HLR method failed to detect. However, all convolutional neural network (CNN)-based methods—EEVD, TSVD_SD, and TSVD_DC—successfully detected the VCF at L2 by extracting and analyzing specific features of spinal images. In TSVD_DC, there was an additional FP, as the model incorrectly identified a VCF at L1.

In Figure 14, according to radiological assessment, the patient had acute VCFs only at the L3 and L5 levels. The T11 vertebra presented a mixed appearance, with both wedge and biconcave features, which could potentially have been interpreted as a VCF. However, it was confirmed to be a normal vertebral body. Despite this, both the HLR and the TSVD_SD, which relied solely on cortical contour information, classified T11 as a VCF, resulting in an FP. For the L3 level, which indeed had an acute VCF, all methodologies accurately detected it as a TP. However, the L5 level was missed by both the HLR and EEVD methods. Due to the relatively uniform height loss, it was challenging for the HLR method to identify, and similarly, the EEVD method produced an FN at L5.

## 4. Discussion

From the confusion matrix results, the combination of HLR(−) + TSVD_SD exhibited the highest sensitivity, making it ideal for applications where identifying positive cases is critical. Conversely, the combination of HLR(+) + EEVD demonstrated the lowest sensitivity, reflecting a conservative classification approach. The combination of HLR(+) + EEVD achieved the highest specificity, making it suitable for minimizing false positives. In contrast, HLR(−) + TSVD_SD demonstrated the lowest specificity due to a higher number of FPs. The TSVD_DC method achieved a good balance between sensitivity and specificity, making it well-suited for general-purpose diagnostic applications.

In sensitivity, specificity, accuracy, and precision results, the TSVD_SD and the HLR(−) + TSVD_SD demonstrated the highest sensitivity (84.46% and 87.84%, respectively), making them suitable for applications where detecting true positives is critical. The combination of HLR(+) + EEVD achieved the highest specificity (99.77%) and precision (97.62%), making it ideal for minimizing false positives. The HLR(−) + TSVD_SD provided a strong balance of sensitivity (87.84%) and specificity (88.89%). Similarly, the combination of HLR(−) + TSVD_DC offered a balanced performance with sensitivity (85.14%) and specificity (89.34%), making it a robust choice for general applications.

When examining the ROC results, the TSVD_SD, with an AUROC of 0.90, outperformed all standalone models, demonstrating its superior ability to distinguish between positive and negative cases. This makes it particularly suitable for scenarios prioritizing the detection of positive cases, such as early disease screening. The HLR(−) + TSVD_SD and HLR(−) + TSVD_DC, both achieving AUROCs of 0.87, demonstrated the highest performance among the combined models. The EEVD, with an AUROC of 0.83, along with its combinations with HLR(−), provided strong specificity, making them suitable for applications where avoiding false positives was essential. The ROC analysis highlighted the superiority of TSVD_SD, both as a standalone model and in combination with HLR(−). These models consistently delivered robust performance across key metrics. In contrast, positive combinations of HLR with other methods yielded mixed results, with HLR(+) + EEVD even performing worse than the baseline model.

Numerous methods have been proposed with similar results in previous literature for the automatic detection of VCFs. However, all of these studies primarily focused on HLR-based Genant grading and did not specifically consider acute VCFs, which was the main motivation of our study (Table 4).

Nadeem S.A. et al. proposed a method for detecting vertebral deformities using vertebral height features and parametric computational modeling [28]. Contour analysis was performed in the central anteroposterior plane, and the anterior-posterior and middle-posterior ratios were calculated as metrics to quantify fracture deformities. In the test dataset, expert readers identified biconcave and wedge-shaped VCFs with moderate or severe deformities in the T1–L1 vertebrae. From 40,050 vertebrae of 3231 chronic obstructive pulmonary disease (COPD) patients, the method demonstrated sensitivity and specificity of 94.8% and 98.5%, respectively. Burns J.E. et al. utilized vertebral height distribution for their analysis [29]. The cross-sectional area of the vertebral body was divided into 17 regions to calculate the overall height distribution of each vertebra. Height patterns were analyzed using support vector regression to differentiate between fractured and normal vertebrae. This method achieved a sensitivity of 95.7% and a false positive rate of 0.29 in a dataset comprising 1275 vertebrae from 150 CT examinations. However, these studies relied solely on vertebral height for diagnosing VCFs and did not consider the classification of acute VCFs.

Studies using commercial software for detecting VCFs have also been conducted. Bendtsen M.G. et al. evaluated the performance of an automated VCF detection software, HealthVCF (Version 5.1.1), which utilized an HLR-based algorithm in a real-world setting at a Danish hospital [30]. Based on the evaluation of 1000 CT scans, the software demonstrated a sensitivity of 0.68 and a specificity of 0.91. The research team concluded that the performance of HealthVCF, which focused solely on the HLR method, was lower than expected and that the tested version lacked generalizability to the Danish population. A similar study conducted by Pereira R.F.B. et al. reported that the HealthVCF software achieved a diagnostic accuracy of 89.6%, a sensitivity of 73.8%, and a specificity of 92.7% across 899 CT scans [31]. These results demonstrated that while the HLR-based algorithm could achieve high specificity, its sensitivity was relatively low. Consequently, integrating deep learning, as proposed in the present study, could potentially enhance detection performance. Page J.H. et al. assessed the diagnostic performance of The Zebra Medical Vision software (the version of this software was not revealed in this reference study), a DL-based VCF detection algorithm, in a study involving 1087 participants of CT images [32]. For moderate-to-severe VCFs, the sensitivity and specificity were reported as 0.78 (95% confidence interval [CI], 0.70–0.85) and 0.87 (95% CI, 0.85–0.89), respectively.

Recent studies have focused on detecting VCFs using DL methods rather than relying on HLR-based approaches. Iyer S. et al. proposed a bounding box-based CNN classification method for the automated detection of VCFs [33]. The research team generated six different 3D bounding boxes, which were divided into 2D sagittal slices centered around the coronal midline. Each slice was further partitioned into patches for CNN classification training to detect VCFs, followed by the application of a majority voting mechanism. The method achieved a sensitivity of 88.10% and a specificity of 84.20% on thoracic CT data from 308 patients. However, this study also classified reference standard VCFs based on fractures defined by the Genant classification, without considering acute VCFs. Dong Q. et al. developed a classifier for determining the Genant grade of each vertebral body using the GoogLeNet architecture [34]. This model achieved a sensitivity of 97.7% and a specificity of 95.1% by maximizing Youden’s J statistic in a dataset of 669 patients. However, the acute or chronic status of the fractures was not considered. For high-performance object detection, Wongthawat et al. utilized the YOLOv8 architecture to detect osteoporotic VCFs [35]. The validation data were classified using the AO Spine-DGOU (German Society for Orthopaedics and Trauma) Osteoporotic Fracture Classification System (OF 1 to OF 5) [36]. This study employed 1050 sagittal CT scan radiographic images of the thoracolumbar region, divided into 934 training cases and 116 test cases. Across grades OF 1 through OF 4, the specificity ranged from 94.28% to 97.86%, while the sensitivity ranged from 95.85% to 97.35%.

HLR-based methods have been widely used as a reference standard in various studies, and significant progress has been made in detecting VCFs based on the Genant classification. However, the findings of this study demonstrated that applying the HLR method directly to detect acute VCFs resulted in very low detection performance. This highlights the necessity for alternative methodologies beyond the HLR approach for identifying clinically significant acute VCFs. The challenges associated with HLR-based detection of VCFs have also been noted in previous studies. Jiang G. et al. reported that height measurement-based approaches can be influenced by subjectivity and may be difficult to apply in certain cases [37]. Furthermore, Lentle B.C. et al. revealed that the HLR assessment method based on the Genant classification exhibited a high rate of inter-observer variability, underscoring its limitations [38]. These findings indicate that while HLR-based methods have been valuable for detecting general VCFs, their limitations must be addressed when applied to the detection of acute VCFs in clinical practice.

Also, the HLR method had a significant limitation in that it could not reliably detect VCFs when key features of acute VCFs in CT, such as cortical step-off, cortical breakage, or cortex discontinuation, were present. Additionally, due to physiological wedging, vertebral bodies classified as normal by radiological standards could still meet the criteria for VCF based on height measurements, leading to FPs. This issue was associated with the size of the vertebral body and represented a limitation of the Genant Classification.

As shown in Figure 15, the five gray trapezoids had identical shapes but were scaled to resemble vertebrae arranged in a comparable manner. Height measurements were conducted on three vertebrae of different sizes. For each vertebra, the height loss was consistently set at 5 units between the anterior and posterior regions, and the HLR value was calculated based on this uniform height loss. When the height loss rate was calculated, the smallest vertebral body exhibited a loss rate of 26.31%, classifying it as a VCF, whereas the largest vertebral body showed a loss rate of 23.81%, allowing it to be classified as normal. The traditional method of evaluating vertebral height loss, without consideration of vertebral shape or size, was subject to variability influenced by vertebral size. Consequently, smaller vertebrae were more likely to be classified as VCFs even with minimal height loss, explaining the higher prevalence of FPs in the thoracic spine compared to the lumbar spine. While it may be reasonable to classify smaller vertebrae with minor height loss as VCFs due to increased risk, these findings highlighted the need for a method that accounts for vertebral size in the quantitative assessment of vertebral compression fractures in the future.

In alignment with recent research trends in VCF detection, our study analyzed the effectiveness of employing DL techniques for the diagnosis of VCFs. For the EEVD, since extensive preprocessing for efficient feature extraction was not performed, the model identified VCFs directly from patient images, similar to how radiologists approach such cases. While EEVD had the advantage of learning information from the surrounding regions of the vertebral body to detect VCFs, it struggled to focus specifically on radiologic characteristics such as cortical bone information, disruptions, or step-offs that are critical for identifying acute VCFs. Although EEVD demonstrated better performance in identifying VCFs compared to the HLR, it occasionally failed to detect clear VCFs. The TSVD_SD method, in contrast to EEVD, relied on highly limited features extracted from the input images for VCF detection. It exclusively learned features from cortical contour information, making it the most suitable method for identifying VCFs caused by morphological changes. The TSVD_DC method, similar to TSVD_SD, attempted to detect VCFs by limiting the information used for analysis. This approach involved cropping only the vertebral region for classification, enabling the detection of VCFs based solely on the vertebral image without the influence of information from surrounding regions.

Previous methodologies did not include studies that performed VCF detection using only vertebral contour information as input. This study demonstrated the utility of a deep learning model trained to detect acute VCFs using solely vertebral contour images. It confirmed the potential of a DL model to detect acute compression fractures even with simple images containing only morphological features of the vertebrae, excluding information from internal vertebral structures, surrounding muscles, or intervertebral discs. The results of this study demonstrated that the TSVD_SD method, which restricted input information to vertebral contour images, effectively captured the clinical characteristics of acute vertebral fractures. This conclusion was supported by the analysis of the ROC curve, as well as sensitivity and specificity results, which highlighted the clinical relevance of this approach in detecting acute fractures.

This study had several limitations. First, this study was conducted using a single-institution, retrospective dataset, which resulted in a relatively small sample size. This limitation may have introduced selection bias and restricted the generalizability of the findings. The challenge in obtaining a fully labeled large-scale dataset for acute VCFs was primarily due to the time-intensive process of labeling clinical VCF data. Despite this limitation, the study provided meaningful insights by investigating clinically classified acute VCFs, even with a small dataset.

Second, specific conditions such as Schmorl’s nodes or other degenerative changes, which can induce morphological alterations similar to vertebral compression fractures (VCFs), were not considered in this study. The exact morphological criteria used to identify vertebrae classified as fractures—particularly the minimum detectable size of fissures or the presence of other early fracture indicators—require further discussion in future studies. Addressing these aspects will be crucial to distinguishing acute VCFs from other conditions with similar morphological features.

In medical imaging studies utilizing deep learning models to detect specific diseases, cross-validation is often employed to address the limitations of small datasets [39]. However, in this study, implementing a random data-splitting approach for cross-validation was challenging due to the distinct data processing methods and structural differences among the three deep learning methodologies. The EEVD model detected VCFs directly from input images, while the TSVD_SD model relied on segmentation results, and the TSVD_DC model used vertebral detection outcomes for classification. These differences made it difficult to ensure consistent performance evaluation across methodologies, even when using the same data-splitting method, as the data flow and dependency on intermediate results varied between models. Additionally, the two-stage models required extra training time for each stage and necessitated rigorous prevention of data leakage, where intermediate results could influence the validation data. This increased the complexity and computational cost of implementing cross-validation. As a result, the application of cross-validation to compare the three deep learning methodologies was restricted due to these inherent characteristics. Nevertheless, a thorough quantitative and qualitative evaluation of the experimental results was conducted using a fixed dataset split. Since the validation dataset remained consistent, it was possible to examine the fixed output results of each methodology and determine which approach was most effective for detecting acute fractures. Additionally, this approach allowed for a detailed analysis of the strengths and weaknesses of each methodology.

Additionally, while this study focused exclusively on the detection of acute VCFs, exploring potential correlations with disc degeneration is equally important. The applicability of the study’s findings should be considered within the broader clinical context, where interactions among various spinal components play a critical role. To enhance the understanding of spinal health, future research should adopt a multidisciplinary approach to investigate the interplay between vertebral and disc health. Such an approach would enable a more holistic evaluation and could potentially lead to the development of more effective treatment strategies.

## 5. Conclusions

This study demonstrated that DL-based methods significantly improved the detection of acute VCFs compared to the traditional HLR methodology. Among the models, the two-stage VCF detection using segmentation and detection, TSVD_SD, exhibited the highest sensitivity and accuracy, making it the most effective standalone approach for identifying VCFs caused by morphological changes. Combining DL methods with HLR further enhanced diagnostic performance, reducing both false positives and false negatives. These findings underscored the limitations of the Genant Classification, particularly its reliance on vertebral height loss measurements without accounting for vertebral size or cortical features. DL approaches such as TSVD_SD and TSVD_DC addressed these limitations by incorporating advanced feature extraction and segmentation techniques.

## Figures and Tables

**Figure 1 bioengineering-12-00064-f001:**
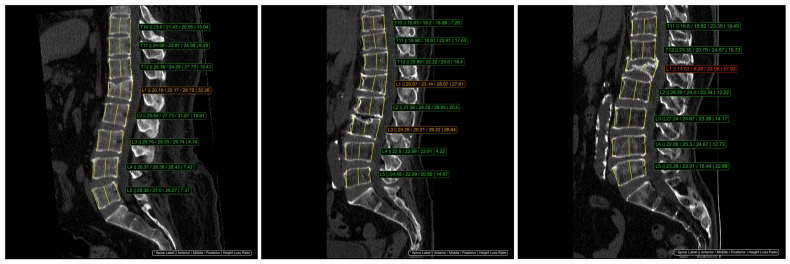
An example of automatic HLR measurement from DL-software, ClariVBA (Ver1.0, ClariPi Inc., Seoul, Republic of Korea). The consecutive values in the image, indicated with an * symbol, were spine label, anterior height, middle height, posterior height, and HLR. Moderate (Genant grade 2) VCF is orange and severe (Genant grade 3) VCF is red. The green color represented the normal vertebrae.

**Figure 2 bioengineering-12-00064-f002:**
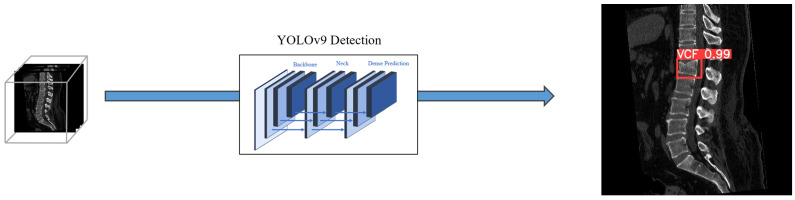
The framework of EEVD method. The YOLOv9 model was utilized to generate bounding boxes on acute VCFs.

**Figure 3 bioengineering-12-00064-f003:**
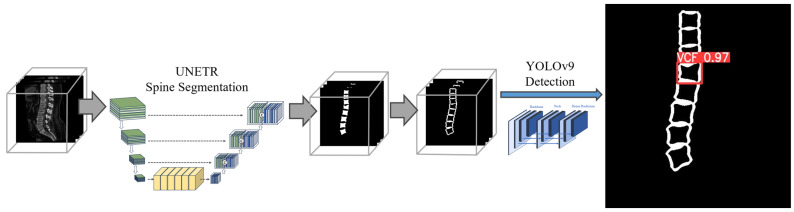
The framework of TSVD_SD method. The contour mask of vertebral body was extracted from segmentation model, UNETR architecture. The YOLOv9 model was used to detect acute VCFs.

**Figure 4 bioengineering-12-00064-f004:**
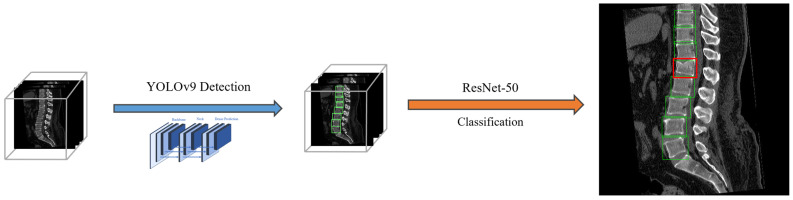
The framework of TSVD_DC method. Bounding boxes of each vertebra were detected and each bounding box was classified by ResNet model. The red bounding box represented the VCF, and the green bounding box represented the normal vertebrae.

**Figure 5 bioengineering-12-00064-f005:**
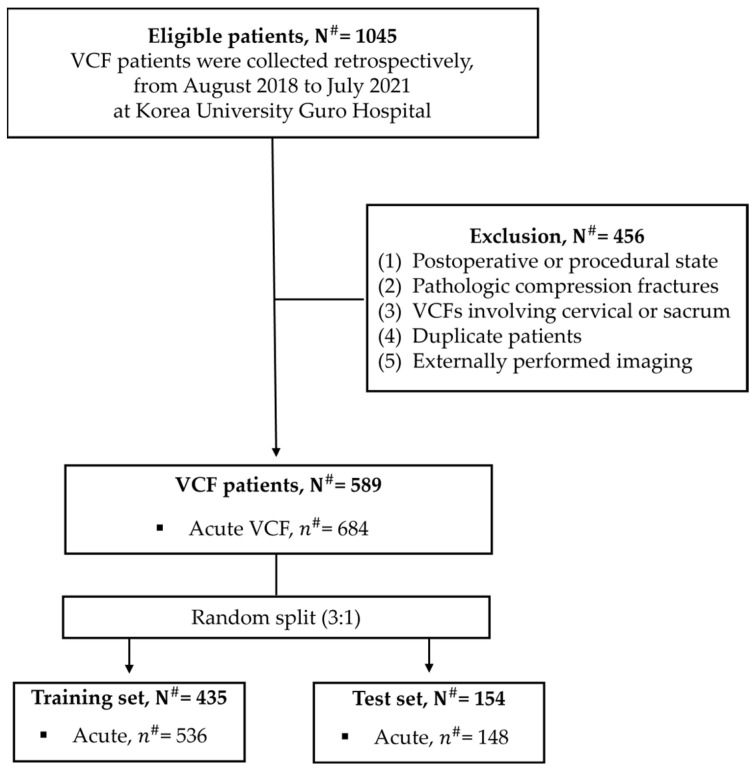
The inclusion criteria and exclusion criteria of this study. N^#^ means the number of patients and *n*^#^ means the number of vertebral bodies from CT images.

**Figure 6 bioengineering-12-00064-f006:**
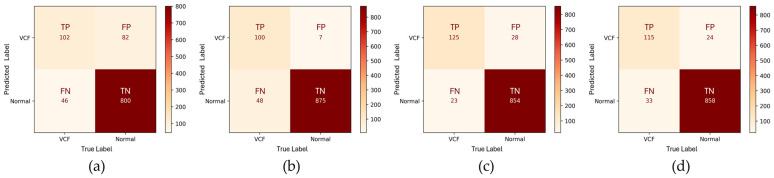
Confusion matrices of four different approaches in evaluation method 1. (**a**) HLR, (**b**) EEVD, (**c**) TSVD_SD, and (**d**) TSVD_DC.

**Figure 7 bioengineering-12-00064-f007:**
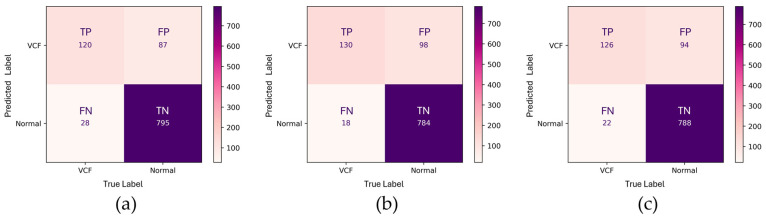
Confusion matrices of three different approaches in evaluation method 2. (**a**) HLR(−) + EEVD, (**b**) HLR(−) + TSVD_SD, and (**c**) HLR(−) + TSVD_DC.

**Figure 8 bioengineering-12-00064-f008:**
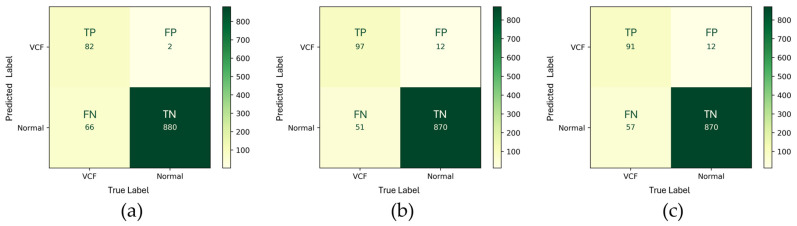
Confusion matrices of three different approaches in evaluation method 3. (**a**) HLR(+) + EEVD, (**b**) HLR(+) + TSVD_SD, and (**c**) HLR(+) + TSVD_DC.

**Figure 9 bioengineering-12-00064-f009:**
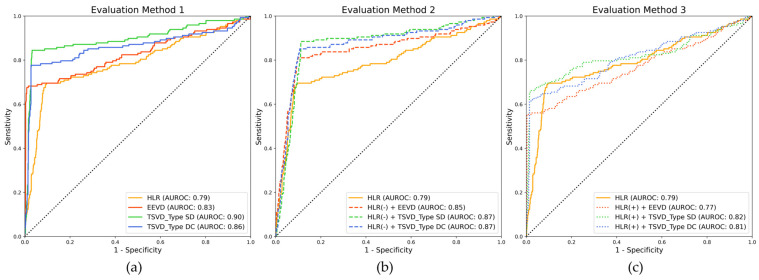
ROC curves of three different evaluation methods. (**a**) Evaluation method 1, (**b**) Evaluation method 2, and (**c**) Evaluation method 3. The yellow solid line means standalone HLR.

**Figure 10 bioengineering-12-00064-f010:**
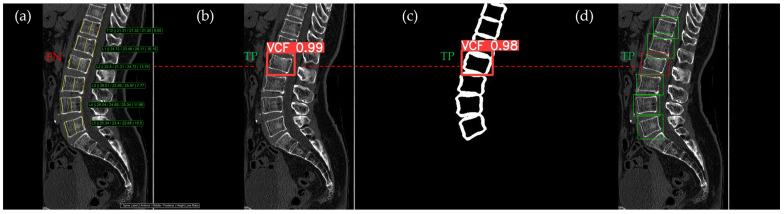
The VCF detection results from four different methods. (**a**) HLR, (**b**) EEVD, (**c**) TSVD_SD, and (**d**) TSVD_DC. The acute VCF, as confirmed by the radiologist, was located at the L2 level. Both the FN and TP were marked at the same level with a red dotted line.

**Figure 11 bioengineering-12-00064-f011:**
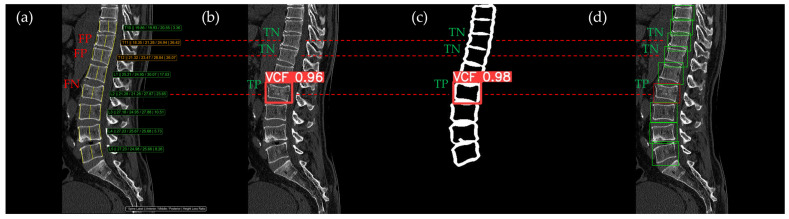
The VCF detection results from four different methods. (**a**) HLR, (**b**) EEVD, (**c**) TSVD_SD, and (**d**) TSVD_DC. The acute VCF, as confirmed by the radiologist, was only located at the L2 level. TP, TN, FP, and FN were marked at the same level with a red dotted line. FP occurred at the thoracic vertebrae only with the HLR.

**Figure 12 bioengineering-12-00064-f012:**
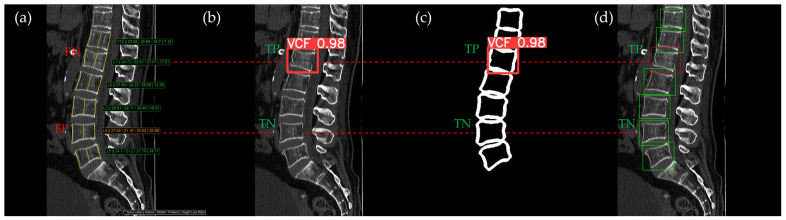
The VCF detection results from four different methods. (**a**) HLR, (**b**) EEVD, (**c**) TSVD_SD, and (**d**) TSVD_DC. The acute VCF, as confirmed by the radiologist, was located at the L1 level. TP, TN, FP, and FN were marked at the same level with a red dotted line.

**Figure 13 bioengineering-12-00064-f013:**
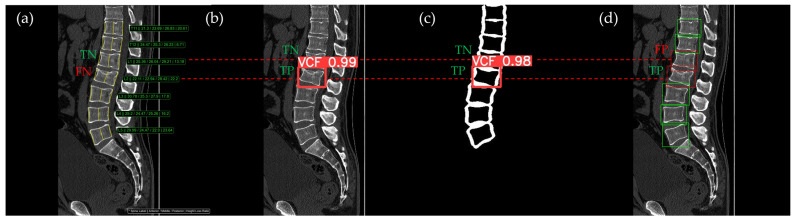
The VCF detection results from four different methods. (**a**) HLR, (**b**) EEVD, (**c**) TSVD_SD, and (**d**) TSVD_DC. The acute VCF, as confirmed by the radiologist, was located at the L2 level. TP, TN, FP, and FN were marked at the same level with a red dotted line. The HLR was low at L2, which was the reason for the FN in (a). The L1 vertebra was misclassified as VCF by TSVD_DC.

**Figure 14 bioengineering-12-00064-f014:**
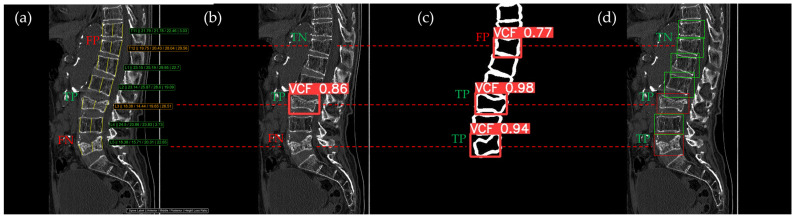
The VCF detection results from four different methods. (**a**) HLR, (**b**) EEVD, (**c**) TSVD_SD, and (**d**) TSVD_DC. The acute VCF, as confirmed by the radiologist, was located at L3 and L5. TP, TN, FP, and FN were marked at the same level with a red dotted line.

**Figure 15 bioengineering-12-00064-f015:**
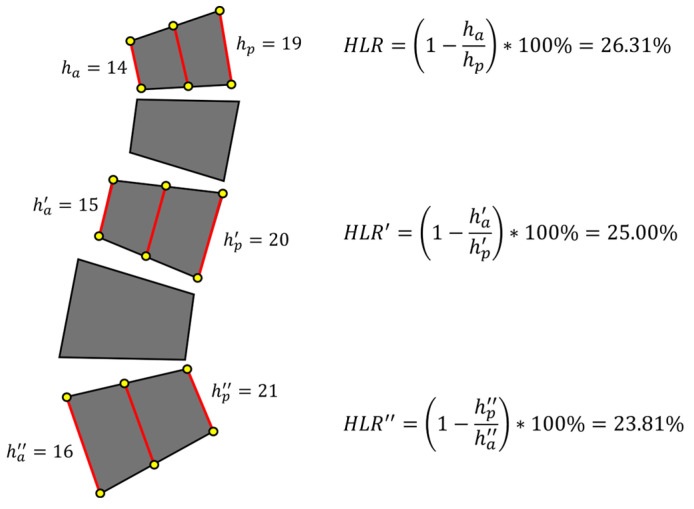
An example illustration explaining the limitations of Genant Classification. Even with identical height loss, the classification of a VCF varies depending on the size of the vertebra. $h_{a}$ means anterior height and $h_{p}$ means posterior height.

**Table 1 bioengineering-12-00064-t001:** Acute VCF detection performance in Evaluation Method 1.

Evaluation Method 1	Sensitivity	Specificity	Accuracy	Precision
HLR	68.92% (61.06–75.82%, 95% CI)	90.70% (88.61–92.45%, 95% CI)	87.57% (85.42–89.45%, 95% CI)	55.43% (48.21–62.43%, 95% CI)
EEVD	67.57% (59.66–74.58%, 95% CI)	99.21% (98.37–99.62%, 95% CI)	94.66% (93.11–95.87%, 95% CI)	93.46% (87.11–96.80%, 95% CI)
TSVD_SD	84.46% (77.76–89.42%, 95% CI)	96.83% (95.45–97.79%, 95% CI)	95.05% (93.55–96.21%, 95% CI)	81.70% (74.82–87.02%, 95% CI)
TSVD_DC	77.70% (70.34–83.66%, 95% CI)	97.28% (95.98–98.16%, 95% CI)	94.47% (92.90–95.70%, 95% CI)	82.73% (75.59–88.11%, 95% CI)

**Table 2 bioengineering-12-00064-t002:** Acute VCF detection performance in Evaluation Method 2.

Evaluation Method 2	Sensitivity	Specificity	Accuracy	Precision
HLR(−) + EEVD	81.08% (74.02–86.57%, 95% CI)	90.14% (87.99–91.93%, 95% CI)	88.83% (86.77–90.62%, 95% CI)	57.97% (51.16–64.49%, 95% CI)
HLR(−) + TSVD_SD	87.84% (81.59–92.17%, 95% CI)	88.89% (86.64–90.80%, 95% CI)	88.74% (86.66–90.53%, 95% CI)	57.02% (50.53–63.27%, 95% CI)
HLR(−) + TSVD_ DC	85.14% (78.52–89.97%, 95% CI)	89.34% (87.13–91.21%, 95% CI)	88.74% (86.66–90.53%, 95% CI)	57.27% (50.67–63.63%, 95% CI)

**Table 3 bioengineering-12-00064-t003:** Acute VCF detection performance in Evaluation Method 3.

Evaluation Method 3	Sensitivity	Specificity	Accuracy	Precision
HLR(+) + EEVD	55.41% (47.36–63.18%, 95% CI)	99.77% (99.18–99.94%, 95% CI)	93.40% (91.71–94.76%, 95% CI)	97.62% (91.73–99.34%, 95% CI)
HLR(+) + TSVD_SD	65.54% (57.58–72.72%, 95% CI)	98.64% (97.64–99.22%, 95% CI)	93.88% (92.25–95.19%, 95% CI)	88.99% (81.74–93.59%, 95% CI)
HLR(+) TSVD_DC	61.49% (53.45–68.94%, 95% CI)	98.64% (97.64–99.22%, 95% CI)	93.30% (91.61–94.67%, 95% CI)	88.35% (80.73–93.21%, 95% CI)

**Table 4 bioengineering-12-00064-t004:** The summary of previous literature for VCF detection.

Author	Key Methodology	Performance Results	VCF Classification Methods
Nadeem S.A. et al. [28]	Quantitative vertebral body contour analysis from 3231 COPD patients.	Sensitivity 94.8% Specificity 98.5%	HLR-based Genant Classification (biconcave and wedge VCFs with Genant grade 2 or 3)
Burns J.E. et al. [29]	Using regression with vertebral height distribution in 17 divided regions from 150 CT examinations.	Sensitivity 95.7% False Positive Rate 0.29	HLR-based Genant Classification (Genant grade 1~3)
Bendtsen M.G. et al. [30]	Commercial VCF detection software, HealthVCF, with 1000 CT scans of real-world setting at Danish hospital.	Sensitivity 0.68 Specificity 0.91	HLR-based Genant Classification
Pereira R.F.B. et al. [31]	HealthVCF with 899 CT scans for moderate-to-severe HLR loss.	Sensitivity 73.8% Specificity 92.7% Accuracy 89.6%	HLR-based Genant Classification (Genant grade 2~3)
Page J.H. et al. [32]	The Zebra Medical Vision software with 1087 participants of CT images.	Sensitivity 0.78 Specificity 0.87	HLR-based Genant Classification
Iyer S. et al. [33]	Bounding-box-based CNN classification for VCF detection with CT data of 308 patients.	Sensitivity 88.1% Specificity 84.2%	HLR-based Genant Classification
Dong Q. et al. [34]	Using GoogLeNet architecture classifier with 669 radiographs.	Sensitivity 97.7% Specificity 95.1%	HLR-based Genant Classification
Wongthawat et al. [35]	YOLOv8 architecture for VCF detection with 116 CT scans.	Specificity 94.28~97.86% Sensitivity 95.85~97.35% (for fracture grade OF 1~4)	AO Spine-DGOU Osteoporotic Fracture Classification System (OF 1 to OF 5)

## Data Availability

The datasets generated or analyzed during the study are available from the corresponding author upon reasonable request.

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
