# Peer review of "Enhanced Detection Performance of Acute Vertebral Compression Fractures Using a Hybrid Deep Learning and Traditional Quantitative Measurement Approach: Beyond the Limitations of Genant Classification"

_bioengineering, 2025, doi:10.3390/bioengineering12010064_

Round 1

Reviewer 1 Report

Comments and Suggestions for Authors

I have examined your study "Enhanced Detection Performance of Acute Vertebral Compression Fractures Using a Hybrid Deep Learning and Traditional Quantitative Measurement Approach: Beyond the Limitations of Genant Classification" in detail. I have listed the points that I found lacking in the article. Eliminating the relevant deficiencies will increase the quality of the article. In the Abstract section, the Objective title should be explained in more detail. In the last paragraph of the Introduction section, the article's contributions to the literature are written. However, the innovative aspects of the article are not included. The last paragraph of the Introduction section should be concluded with a paragraph that includes the article's organization. The Proposed Methods section should be written in more detail, and an explanatory figure about the model should be added. The relevant section gives very general information about the models in the literature. The model you proposed should be highlighted. The application results are presented in detail. However, the organization should be reviewed. When the confusion matrices are examined, I do not think there is a need for too much detailed explanation. The words "we, our etc." should be avoided. Spelling and grammatical errors should be reviewed.

Author Response

Comments 1:

In the Abstract section, the Objective title should be explained in more detail.

Response 1:

We appreciate the reviewer's concern regarding the need to explain more detail objective in Abstract section.

Based on your suggestion, we revised our objective part at Abstract section.

“Objective:

This study evaluated the applicability of the classical method, height loss ratio (HLR), for identifying major acute compression fractures in clinical practice and compared its perfor-mance with deep learning (DL)-based VCF detection methods. Additionally, it examined whether combining the HLR with DL approaches could enhance performance, exploring the po-tential integration of classical and DL methodologies.”

Comments 2: 

In the last paragraph of the Introduction section, the article's contributions to the literature are written.

However, the innovative aspects of the article are not included.

The last paragraph of the Introduction section should be concluded with a paragraph that includes the article's organization.

Response 2:

Thank you for your thoughtful comments.

Based on your suggestion, we revised last part of Introduction section.

We added the comments in the “1.4 Purpose and Proposed Approaches” section.

“1.4. Purpose and Proposed Approaches

This study evaluated the applicability of the HLR, traditionally limited to assessing vertical spinal height deformation, for detecting major acute compression fractures in clinical settings, while identifying its limitations. To enhance the detection performance of acute VCFs, a novel DL-based method was proposed that utilized the full structural in-formation of the vertebral body, rather than considering only vertical deformations. This approach aimed to demonstrate the potential of DL in detecting acute VCFs. As part of this investigation, a DL model based on vertebral contour information—a concept not previ-ously explored—was introduced to examine its utility in detecting fractures, including horizontal deformations. This model highlighted the applicability of DL in addressing the limitations of traditional quantitative HLR methods. Finally, an integrated DL detection process was proposed, combining HLR-based measurements with the newly developed DL model to overcome the limitations of existing approaches.

To achieve these objectives, the commercial software, ClariVBA (Ver1.0, ClariPi Inc., Seoul, Republic of Korea), for calculating vertebral HLR and newly developed DL tech-niques was implemented. The respective performances of these methods in detecting acute VCFs were then evaluated. Additionally, the detection performance of the combined approach, integrating HLR measurements with DL methods, was compared and assessed. For the DL-based detection of acute VCFs, three distinct methodologies were applied. These included: an end-to-end model for directly detecting fractures from input images, a two-stage method that first segmented vertebral body contours and subsequently detected fractures within the segmented regions, and another two-stage model that identified indi-vidual vertebral bodies and classified them as either having VCFs or not. A comparative analysis of these methodologies was conducted to identify the most effective DL approach for this purpose. This structured investigation emphasized the potential of integrating traditional quantitative measurements with advanced DL techniques to improve the de-tection and evaluation of acute VCFs.”

Comments 3: 

The Proposed Methods section should be written in more detail, and an explanatory figure about the model should be added.

The relevant section gives very general information about the models in the literature.

The model you proposed should be highlighted.

Response 3:

Thank you for your constructive comments.

We agreed with the concerns regarding the detailed explanation of our model.

Based on your suggestion, we added more comments at “2.1. Proposed Methods” section.

Also, we added explanatory figures, Figure 1~4, at “2.1. Proposed Methods” section.

“The height of three regions measured for each vertebral body was calculated accord-ing to the Genant Classification method, categorized into four groups: normal (HLR < 20%), mild (20% ≤ HLR < 25%), moderate (25% ≤ HLR < 40%), and severe (HLR ≥ 40%) [6]. The ClariVBA program automatically measured the height of the three regions for each vertebral body, calculated the height loss ratio, and provided quantitative values adjacent to each vertebra. Moderate fractures were displayed in orange, severe fractures in red, and mild fractures, which could exhibit inter- or intra-observer variation, were displayed in the same green color as normal vertebrae [Figure 1]. In this study, vertebrae with mild com-pression were not classified as VCFs. “

“A model designed to directly detect VCFs from input images using DL was regarded as the simplest approach for VCF detection. This method trained a DL model to identify VCFs directly from input images without requiring complex processing steps, mimicking the way a human expert would analyze the images. By focusing solely on determining the location of VCFs in the input images, the utility of the trained DL model was evaluated. “

“This method focused on detecting VCFs by emphasizing vertebral contour information, which is a key feature of acute VCFs, through vertebral segmentation. The model was de-signed to restrict other information outside the vertebral contour, enabling detection based on the irregularities in the vertebral contour regions. Although features from the sur-rounding vertebral areas or internal vertebral structures could potentially aid the model's training process, the morphological deformation of the vertebral contour, as observed in CT images, was identified as the critical characteristic of acute VCFs, leading to the de-velopment of this model. “

“By pre-identifying the location of each vertebra using bounding boxes, it was possible to evaluate the presence of VCFs while considering only the individual vertebral regions. This approach enabled the development of a classification model that could learn not only the morphological features of vertebrae but also the characteristics of the trabecular bone region within the vertebrae by training on cropped images of individual vertebrae. “

Comments 4: 

The application results are presented in detail. However, the organization should be reviewed.

When the confusion matrices are examined, I do not think there is a need for too much detailed explanation.

Response 4:

We appreciate the reviewer's concern regarding the need to organize “3.Result” section.

Based on your suggestion, we revised our “3.Result” section to avoid redundancy and convey the meaning concisely.

Comments 5: 

The words "we, our etc." should be avoided.

Spelling and grammatical errors should be reviewed.

Response 5:

Thank you for your thoughtful comments.

Following your suggestion, we tried to avoid using the word "we, our”.

Additionally, we checked for grammatical and spelling errors.

Reviewer 2 Report

Comments and Suggestions for Authors

The work is interesting, the obtained results are interesting. However, a number of issues require explanation before the work can be published.

 The paper lacks a literature review. The authors should perform it and then identify gaps in the current state of the art. Then it should be shown how the conducted research will fill these gaps. Next, it will it be possible to assess the contribution of this work to the field and its novelty.

The authors used a division into a test and training set to validate the developed DL models. However, in this case, the five-fold cross-validation method would be more reliable approach. Please apply this validation method to at least the most promising models.

The discussion shows few comparisons of the obtained results with similar results already published by other researchers. More such cases should be provided and presented in the form of a table for greater readability. The discussion also lacks a clear conclusion as to which model is the best from the point of view of clinical applications and whether the achieved values of the model quality measures are sufficient for such applications.

Author Response

Thank you for taking the time to review this manuscript with constructive comments.

We have revised the manuscript as you and other reviewers have suggested.

The responses to the comments are as follows.

Comments 1:

The paper lacks a literature review.

The authors should perform it and then identify gaps in the current state of the art.

Then it should be shown how the conducted research will fill these gaps.

Next, it will it be possible to assess the contribution of this work to the field and its novelty.

Response 1:

We appreciate the reviewer's concern regarding the lacks of literature review and assessments of this study.

We agree with the reviewer’s opinion that we have to perform literature reviews and then identify gaps to assess the contribution of this work to the field and its novelty.

We added the “1.3. Related Works” and “1.4. Purpose and Proposed Approaches” parts at 1.Introduction section.

“1.3. Related Works

Numerous studies had been conducted to develop DL models based on radiographs, plain radiographs were typically used as the first-line investigation for diagnosing VCFs. However, acute VCFs were not specifically considered in these studies.

Cheng L. et al. detected each vertebra on pre-processed lateral spine X-ray images and segmented each vertebral body solely for the classification of normal vertebrae, compres-sion fractures, and burst fractures [15]. This study utilized the YOLOv4 and ResUNet models, achieving a precision of 74% in identifying compression fractures; however, the acute state of the fractures was not considered. Seo J. W. et al. developed an automated method for measuring vertebral body compression exclusively in the vertical direction from X-ray images using segmentation and regression based on a convolutional neural network (CNN) [16]. This study focused solely on the vertical compression of the vertebral body. Murata K. et al. introduced a deep convolutional neural network (DCNN) using the Visual Recognition V3 model to classify VCF patients from plain thoracolumbar radiog-raphy [17]. Their model achieved 84.7% sensitivity and 87.3% specificity in a dataset of 300 patients; however, the acute state of the fractures was not specified. Guermazi A. et al. assessed the impact of DL model, based on Detectron2, on the diagnostic performance of physicians [18]. Although this study exclusively considered acute fractures, the dataset of 480 radiographs included various body parts beyond the spine, such as the ankle, knee, leg, hip, wrist, and elbow. Chen H. Y. et al. employed a ResNet-based DL model to evalu-ate the feasibility of detecting and localizing VCFs from plain abdominal frontal radio-graphs. Also, this study was limited to the Genant classification method [19].

Baum T. et al. investigated the feasibility of using the anterior-posterior ratio and middle-posterior ratio for diagnosing osteoporotic vertebral fractures, utilizing only the Genant classification method [20]. Tomita N. et al. employed CNN and long short-term memory (LSTM) networks to detect VCFs on CT examinations [21]. All 129 CT scans in the test set were established as reference standards using the Genant classification. Li Y. et al. focused on evaluating benign and malignant vertebral fractures using a ResNet architec-ture [22]. Osteoporotic compression fractures were categorized as benign fractures, whereas malignant fractures were associated with metastatic cancer. However, the acute state of the fractures was not identified. Also, acute VCFs were not specifically considered in previous studies with CT images.

Clinically, detecting acute VCF is crucial [8]. However, most previous studies focused solely on classifying VCFs based on the Genant grade, which relies on height compression ratios, and exploring methodologies for this classification. It remains necessary to verify whether the existing height loss measurement methods are effective for detecting the acute compression fractures that are clinically significant. Developing new approaches for de-tecting acute compression fractures represents an unmet need in the current research landscape.”

“1.4. Purpose and Proposed Approaches

This study evaluated the applicability of the HLR, traditionally limited to assessing vertical spinal height deformation, for detecting major acute compression fractures in clinical settings, while identifying its limitations. To enhance the detection performance of acute VCFs, a novel DL-based method was proposed that utilized the full structural in-formation of the vertebral body, rather than considering only vertical deformations. This approach aimed to demonstrate the potential of DL in detecting acute VCFs. As part of this investigation, a DL model based on vertebral contour information—a concept not previ-ously explored—was introduced to examine its utility in detecting fractures, including horizontal deformations. This model highlighted the applicability of DL in addressing the limitations of traditional quantitative HLR methods. Finally, an integrated DL detection process was proposed, combining HLR-based measurements with the newly developed DL model to overcome the limitations of existing approaches.

To achieve these objectives, the commercial software, ClariVBA (Ver1.0, ClariPi Inc., Seoul, Republic of Korea), for calculating vertebral HLR and newly developed DL tech-niques was implemented. The respective performances of these methods in detecting acute VCFs were then evaluated. Additionally, the detection performance of the combined approach, integrating HLR measurements with DL methods, was compared and assessed. For the DL-based detection of acute VCFs, three distinct methodologies were applied. These included: an end-to-end model for directly detecting fractures from input images, a two-stage method that first segmented vertebral body contours and subsequently detected fractures within the segmented regions, and another two-stage model that identified indi-vidual vertebral bodies and classified them as either having VCFs or not. A comparative analysis of these methodologies was conducted to identify the most effective DL approach for this purpose. This structured investigation emphasized the potential of integrating traditional quantitative measurements with advanced DL techniques to improve the de-tection and evaluation of acute VCFs.”

Comments 2: 

The authors used a division into a test and training set to validate the developed DL models.

However, in this case, the five-fold cross-validation method would be more reliable approach.

Please apply this validation method to at least the most promising models.

Response 2:

Thank you for your constructive comments.

We agreed with the concerns regarding the use of cross-validation.

We added this limitation in our “4.Discussion” section.

In this study, however, implementing a random data-splitting approach for cross-validation was challenging due to the distinct data processing methods and structural differences among the three deep learning methodologies. The EEVD model detected VCFs directly from input images, while the TSVD_SD model relied on segmentation results, and the TSVD_DC model used vertebral detection outcomes for classification. These differences made it difficult to ensure consistent performance evaluation across methodologies, even when using the same data-splitting method, as the data flow and dependency on intermediate results varied between models. Additionally, the two stage models required extra training time for each stage and necessitated rigorous prevention of data leakage, where intermediate results could influence the validation data. This increased the complexity and computational cost of implementing cross-validation. As a result, the application of cross-validation to compare the three deep learning methodologies was restricted due to these inherent characteristics. Nevertheless, a thorough quantitative and qualitative evaluation of the experimental results was conducted using a fixed dataset split. Since the validation dataset remained consistent, it was possible to examine the fixed output results of each methodology and determine which approach was most effective for detecting acute fractures. Additionally, this approach allowed for a detailed analysis of the strengths and weaknesses of each methodology.

Also, this study aimed to identify the most effective deep learning methodology for detecting acute VCFs. Another objective was to evaluate whether combining the height loss ratio method with DL could improve detection performance compared to using each method independently. Consequently, selecting a specific methodology for cross-validation and assessing its optimal performance will be a critical objective for future studies. These subsequent investigations will aim to apply the chosen approach to larger datasets to further enhance its clinical applicability and generalizability.

Comments 3: 

The discussion shows few comparisons of the obtained results with similar results already published by other researchers.

More such cases should be provided and presented in the form of a table for greater readability.

The discussion also lacks a clear conclusion as to which model is the best from the point of view of clinical applications and whether the achieved values of the model quality measures are sufficient for such applications.

Response 3:

Thank you for your insightful comments.

To address the reviewer’s comments, we have incorporated additional comparisons of the obtained results with previously published studies.

These comparisons are presented in [Table 4] to enhance readability and provide a clearer understanding of the relative performance of the proposed models.

Furthermore, the “4.Discussion” has been expanded to include a detailed analysis of the clinical applicability, highlighting the most suitable approach based on the observed results and relevant clinical thresholds.

We added numerous review of related previous literatures for “4. Discussion” section with [Table 4].

“Nadeem S.A. et al. proposed a method for detecting vertebral deformities using verte-bral height features and parametric computational modeling [28]. Contour analysis was performed in the central anteroposterior plane, and the anterior-posterior and mid-dle-posterior ratios were calculated as metrics to quantify fracture deformities. In the test dataset, expert readers identified biconcave and wedge-shaped VCFs with moderate or severe deformities in the T1-L1 vertebrae. From 40,050 vertebrae of 3,231 chronic obstruc-tive pulmonary disease (COPD) patients, the method demonstrated sensitivity and speci-ficity of 94.8% and 98.5%, respectively. Burns J.E. et al. utilized vertebral height distribu-tion for their analysis [29]. The cross-sectional area of the vertebral body was divided into 17 regions to calculate the overall height distribution of each vertebra. Height patterns were analyzed using support vector regression to differentiate between fractured and normal vertebrae. This method achieved a sensitivity of 95.7% and a false positive rate of 0.29 in a dataset comprising 1,275 vertebrae from 150 CT examinations. However, these studies relied solely on vertebral height for diagnosing VCFs and did not consider the classification of acute VCFs.

Studies using commercial software for detecting VCFs have also been conducted. Bendtsen M.G. et al. evaluated the performance of an automated VCF detection software, HealthVCF (Version 5.1.1), which utilized a HLR-based algorithm in a real-world setting at a Danish hospital [30]. Based on the evaluation of 1,000 CT scans, the software demon-strated a sensitivity of 0.68 and a specificity of 0.91. The research team concluded that the performance of HealthVCF, which focused solely on the HLR method, was lower than ex-pected and that the tested version lacked generalizability to the Danish population. A similar study conducted by Pereira R.F.B. et al. reported that the HealthVCF software achieved a diagnostic accuracy of 89.6%, a sensitivity of 73.8%, and a specificity of 92.7% across 899 CT scans [31]. These results demonstrated that while the HLR-based algorithm could achieve high specificity, its sensitivity was relatively low. Consequently, integrating deep learning, as proposed in the present study, could potentially enhance detection per-formance. Page J.H. et al. assessed the diagnostic performance of The Zebra Medical Vi-sion software (the version of this software was not revealed in this reference study), a DL-based VCF detection algorithm, in a study involving 1,087 participants of CT images [32]. For moderate to severe VCFs, the sensitivity and specificity were reported as 0.78 (95% confidence interval [CI], 0.70–0.85) and 0.87 (95% CI, 0.85–0.89), respectively.

Recent studies have focused on detecting VCFs using DL methods rather than relying on HLR-based approaches. Iyer S. et al. proposed a bounding box-based CNN classifica-tion method for the automated detection of VCFs [33]. The research team generated six different 3D bounding boxes, which were divided into 2D sagittal slices centered around the coronal midline. Each slice was further partitioned into patches for CNN classification training to detect VCFs, followed by the application of a majority voting mechanism. The method achieved a sensitivity of 88.10% and a specificity of 84.20% on thoracic CT data from 308 patients. However, this study also classified reference standard VCFs based on fractures defined by the Genant classification, without considering acute VCFs. Dong Q. et al. developed a classifier for determining the Genant grade of each vertebral body using the GoogLeNet architecture [34]. This model achieved a sensitivity of 97.7% and a speci-ficity of 95.1% by maximizing Youden’s J statistic in a dataset of 669 patients. However, the acute or chronic status of the fractures was not considered. “

“Previous methodologies did not include studies that performed VCF detection using only vertebral contour information as input. This study demonstrated the utility of a deep learning model trained to detect acute VCFs using solely vertebral contour images. It con-firmed the potential of a DL model to detect acute compression fractures even with simple images containing only morphological features of the vertebrae, excluding information from internal vertebral structures, surrounding muscles, or intervertebral discs. The re-sults of this study demonstrated that the TSVD_SD method, which restricted input infor-mation to vertebral contour images, effectively captured the clinical characteristics of acute vertebral fractures. This conclusion was supported by the analysis of the ROC curve, as well as sensitivity and specificity results, which highlighted the clinical relevance of this approach in detecting acute fractures.“

Reviewer 3 Report

Comments and Suggestions for Authors

The manuscript is devoted to the comparison of deep learning models for computer tomography image analysis. This problem fully corresponds to the aims and scope of "Bioengineering".

There are the following comments:

1. At the end of Introduction, it is necessary to add a description of the further structure of the manuscript.

2. The literature review is completely formal and needs improvement. References such as [14-23], like in line 57, without discussing the results obtained are not allowed. Section "Related Works" is needed.

3. Authors should add a point-by-point description of how this research contributes to the subject area.

4. The overall framework of the methodology should be presented, including a graphical representation.

5. In the text of the manuscript, before presenting the methodology, it is necessary to provide a formal statement of the problem that needs to be solved.

6. What is the scientific novelty of this manuscript? The methods proposed are neither unique nor developed by the authors of the paper.

7. The dataset is private and, therefore, the reproducibility of the results raises questions. Authors should provide metrics for an openly available dataset. Additionally, the manuscript fails to include a comparison between this solution and other architectures. Section 4 is, therefore, more appropriate for a review/introduction, as it does not compare the results directly.

Author Response

Thank you for taking the time to review this manuscript with constructive comments.

We have revised the manuscript as you and other reviewers have suggested.

The responses to the comments are as follows.

Comments 1:

At the end of Introduction, it is necessary to add a description of the further structure of the manuscript.

Response 1:

Thank you for your thoughtful comments.

Based on your suggestion, we revised last part of Introduction section.

We added the comments in the “1.4 Purpose and Proposed Approaches” section.

“1.4. Purpose and Proposed Approaches

To achieve these objectives, the commercial software, ClariVBA (Ver1.0, ClariPi Inc., Seoul, Republic of Korea), for calculating vertebral HLR and newly developed DL tech-niques was implemented. The respective performances of these methods in detecting acute VCFs were then evaluated. Additionally, the detection performance of the combined approach, integrating HLR measurements with DL methods, was compared and assessed. For the DL-based detection of acute VCFs, three distinct methodologies were applied. These included: an end-to-end model for directly detecting fractures from input images, a two-stage method that first segmented vertebral body contours and subsequently detected fractures within the segmented regions, and another two-stage model that identified indi-vidual vertebral bodies and classified them as either having VCFs or not. A comparative analysis of these methodologies was conducted to identify the most effective DL approach for this purpose. This structured investigation emphasized the potential of integrating traditional quantitative measurements with advanced DL techniques to improve the de-tection and evaluation of acute VCFs.”

Comments 2: 

The literature review is completely formal and needs improvement. References such as [14-23], like in line 57, without discussing the results obtained are not allowed. Section "Related Works" is needed.

Response 2:

We appreciate the reviewer's concern regarding the lacks of literature review.

We added the “1.3. Related Works” parts at 1.Introduction section.

“1.3. Related Works

Numerous studies had been conducted to develop DL models based on radiographs, plain radiographs were typically used as the first-line investigation for diagnosing VCFs. However, acute VCFs were not specifically considered in these studies.

Cheng L. et al. detected each vertebra on pre-processed lateral spine X-ray images and segmented each vertebral body solely for the classification of normal vertebrae, compres-sion fractures, and burst fractures [15]. This study utilized the YOLOv4 and ResUNet models, achieving a precision of 74% in identifying compression fractures; however, the acute state of the fractures was not considered. Seo J. W. et al. developed an automated method for measuring vertebral body compression exclusively in the vertical direction from X-ray images using segmentation and regression based on a convolutional neural network (CNN) [16]. This study focused solely on the vertical compression of the vertebral body. Murata K. et al. introduced a deep convolutional neural network (DCNN) using the Visual Recognition V3 model to classify VCF patients from plain thoracolumbar radiog-raphy [17]. Their model achieved 84.7% sensitivity and 87.3% specificity in a dataset of 300 patients; however, the acute state of the fractures was not specified. Guermazi A. et al. assessed the impact of DL model, based on Detectron2, on the diagnostic performance of physicians [18]. Although this study exclusively considered acute fractures, the dataset of 480 radiographs included various body parts beyond the spine, such as the ankle, knee, leg, hip, wrist, and elbow. Chen H. Y. et al. employed a ResNet-based DL model to evalu-ate the feasibility of detecting and localizing VCFs from plain abdominal frontal radio-graphs. Also, this study was limited to the Genant classification method [19].

Baum T. et al. investigated the feasibility of using the anterior-posterior ratio and middle-posterior ratio for diagnosing osteoporotic vertebral fractures, utilizing only the Genant classification method [20]. Tomita N. et al. employed CNN and long short-term memory (LSTM) networks to detect VCFs on CT examinations [21]. All 129 CT scans in the test set were established as reference standards using the Genant classification. Li Y. et al. focused on evaluating benign and malignant vertebral fractures using a ResNet architec-ture [22]. Osteoporotic compression fractures were categorized as benign fractures, whereas malignant fractures were associated with metastatic cancer. However, the acute state of the fractures was not identified. Also, acute VCFs were not specifically considered in previous studies with CT images.

Clinically, detecting acute VCF is crucial [8]. However, most previous studies focused solely on classifying VCFs based on the Genant grade, which relies on height compression ratios, and exploring methodologies for this classification. It remains necessary to verify whether the existing height loss measurement methods are effective for detecting the acute compression fractures that are clinically significant. Developing new approaches for de-tecting acute compression fractures represents an unmet need in the current research landscape.”

Comments 3:

Authors should add a point-by-point description of how this research contributes to the subject area.

Response 3:

Thank you for your constructive comments.

We added an explanation of how this study contributes to the field at “4.Discussion” section.

“Our study analyzed the effectiveness of employing DL techniques for the diagnosis of VCFs. For the EEVD, since extensive preprocessing for efficient feature extraction was not performed, the model identified VCFs directly from patient images, similar to how radi-ologists approach such cases. While EEVD had the advantage of learning information from the surrounding regions of the vertebral body to detect VCFs, it struggled to focus specifically on radiologic characteristics such as cortical bone information, disruptions, or step-offs that are critical for identifying acute VCFs. Although EEVD demonstrated better performance in identifying VCFs compared to the HLR, it occasionally failed to detect clear VCFs. The TSVD_SD method, in contrast to EEVD, relied on highly limited features extracted from the input images for VCF detection. It exclusively learned features from cor-tical contour information, making it the most suitable method for identifying VCFs caused by morphological changes. The TSVD_DC method, similar to TSVD_SD, attempted to de-tect VCFs by limiting the information used for analysis. This approach involved cropping only the vertebral region for classification, enabling the detection of VCFs based solely on the vertebral image without the influence of information from surrounding regions.

Previous methodologies did not include studies that performed VCF detection using only vertebral contour information as input. This study demonstrated the utility of a deep learning model trained to detect acute VCFs using solely vertebral contour images. It con-firmed the potential of a DL model to detect acute compression fractures even with simple images containing only morphological features of the vertebrae, excluding information from internal vertebral structures, surrounding muscles, or intervertebral discs. The re-sults of this study demonstrated that the TSVD_SD method, which restricted input infor-mation to vertebral contour images, effectively captured the clinical characteristics of acute vertebral fractures. This conclusion was supported by the analysis of the ROC curve, as well as sensitivity and specificity results, which highlighted the clinical relevance of this approach in detecting acute fractures.”

“a thorough quantitative and qualitative evaluation of the experimental results was con-ducted using a fixed dataset split. Since the validation dataset remained consistent, it was possible to examine the fixed output results of each methodology and determine which approach was most effective for detecting acute fractures. Additionally, this approach al-lowed for a detailed analysis of the strengths and weaknesses of each methodology.”

Comments 4: 

The overall framework of the methodology should be presented, including a graphical representation.

Response 4:

Thank you for your constructive comments.

We agreed with the concerns regarding the graphical representation of our methodology.

We added explanatory figures, Figure 1~4, at “2.1. Proposed Methods” section.

Also, we added more comments at “2.1. Proposed Methods” section.

“The height of three regions measured for each vertebral body was calculated accord-ing to the Genant Classification method, categorized into four groups: normal (HLR < 20%), mild (20% ≤ HLR < 25%), moderate (25% ≤ HLR < 40%), and severe (HLR ≥ 40%) [6]. The ClariVBA program automatically measured the height of the three regions for each vertebral body, calculated the height loss ratio, and provided quantitative values adjacent to each vertebra. Moderate fractures were displayed in orange, severe fractures in red, and mild fractures, which could exhibit inter- or intra-observer variation, were displayed in the same green color as normal vertebrae [Figure 1]. In this study, vertebrae with mild com-pression were not classified as VCFs. “

“A model designed to directly detect VCFs from input images using DL was regarded as the simplest approach for VCF detection. This method trained a DL model to identify VCFs directly from input images without requiring complex processing steps, mimicking the way a human expert would analyze the images. By focusing solely on determining the location of VCFs in the input images, the utility of the trained DL model was evaluated. “

“This method focused on detecting VCFs by emphasizing vertebral contour information, which is a key feature of acute VCFs, through vertebral segmentation. The model was de-signed to restrict other information outside the vertebral contour, enabling detection based on the irregularities in the vertebral contour regions. Although features from the sur-rounding vertebral areas or internal vertebral structures could potentially aid the model's training process, the morphological deformation of the vertebral contour, as observed in CT images, was identified as the critical characteristic of acute VCFs, leading to the de-velopment of this model. “

“By pre-identifying the location of each vertebra using bounding boxes, it was possible to evaluate the presence of VCFs while considering only the individual vertebral regions. This approach enabled the development of a classification model that could learn not only the morphological features of vertebrae but also the characteristics of the trabecular bone region within the vertebrae by training on cropped images of individual vertebrae. “

Comments 5: 

In the text of the manuscript, before presenting the methodology, it is necessary to provide a formal statement of the problem that needs to be solved.

Response 5:

We appreciate the reviewer's concern regarding the formal statement of the problem that needs to be solved

Based on your suggestion, we revised last part of Introduction section.

We added the comments in the “1.4 Purpose and Proposed Approaches” section.

“1.4. Purpose and Proposed Approaches

This study evaluated the applicability of the HLR, traditionally limited to assessing vertical spinal height deformation, for detecting major acute compression fractures in clinical settings, while identifying its limitations. To enhance the detection performance of acute VCFs, a novel DL-based method was proposed that utilized the full structural in-formation of the vertebral body, rather than considering only vertical deformations. This approach aimed to demonstrate the potential of DL in detecting acute VCFs. As part of this investigation, a DL model based on vertebral contour information—a concept not previ-ously explored—was introduced to examine its utility in detecting fractures, including horizontal deformations. This model highlighted the applicability of DL in addressing the limitations of traditional quantitative HLR methods. Finally, an integrated DL detection process was proposed, combining HLR-based measurements with the newly developed DL model to overcome the limitations of existing approaches.”

Comments 6: 

What is the scientific novelty of this manuscript? The methods proposed are neither unique nor developed by the authors of the paper.

Response 6:

Thank you for your insightful comments.

We agreed with the concerns regarding the explanation of scientific novelty and differential points.

To address the reviewer’s comments, we have incorporated additional comparisons of the obtained results with previously published studies with [Table 4].

We added more comments for scientific novelty at the “4.Discussion” section, including a detailed analysis of the clinical applicability.

“Nadeem S.A. et al. proposed a method for detecting vertebral deformities using verte-bral height features and parametric computational modeling [28]. Contour analysis was performed in the central anteroposterior plane, and the anterior-posterior and mid-dle-posterior ratios were calculated as metrics to quantify fracture deformities. In the test dataset, expert readers identified biconcave and wedge-shaped VCFs with moderate or severe deformities in the T1-L1 vertebrae. From 40,050 vertebrae of 3,231 chronic obstruc-tive pulmonary disease (COPD) patients, the method demonstrated sensitivity and speci-ficity of 94.8% and 98.5%, respectively. Burns J.E. et al. utilized vertebral height distribu-tion for their analysis [29]. The cross-sectional area of the vertebral body was divided into 17 regions to calculate the overall height distribution of each vertebra. Height patterns were analyzed using support vector regression to differentiate between fractured and normal vertebrae. This method achieved a sensitivity of 95.7% and a false positive rate of 0.29 in a dataset comprising 1,275 vertebrae from 150 CT examinations. However, these studies relied solely on vertebral height for diagnosing VCFs and did not consider the classification of acute VCFs.

Studies using commercial software for detecting VCFs have also been conducted. Bendtsen M.G. et al. evaluated the performance of an automated VCF detection software, HealthVCF (Version 5.1.1), which utilized a HLR-based algorithm in a real-world setting at a Danish hospital [30]. Based on the evaluation of 1,000 CT scans, the software demon-strated a sensitivity of 0.68 and a specificity of 0.91. The research team concluded that the performance of HealthVCF, which focused solely on the HLR method, was lower than ex-pected and that the tested version lacked generalizability to the Danish population. A similar study conducted by Pereira R.F.B. et al. reported that the HealthVCF software achieved a diagnostic accuracy of 89.6%, a sensitivity of 73.8%, and a specificity of 92.7% across 899 CT scans [31]. These results demonstrated that while the HLR-based algorithm could achieve high specificity, its sensitivity was relatively low. Consequently, integrating deep learning, as proposed in the present study, could potentially enhance detection per-formance. Page J.H. et al. assessed the diagnostic performance of The Zebra Medical Vi-sion software (the version of this software was not revealed in this reference study), a DL-based VCF detection algorithm, in a study involving 1,087 participants of CT images [32]. For moderate to severe VCFs, the sensitivity and specificity were reported as 0.78 (95% confidence interval [CI], 0.70–0.85) and 0.87 (95% CI, 0.85–0.89), respectively.

Recent studies have focused on detecting VCFs using DL methods rather than relying on HLR-based approaches. Iyer S. et al. proposed a bounding box-based CNN classifica-tion method for the automated detection of VCFs [33]. The research team generated six different 3D bounding boxes, which were divided into 2D sagittal slices centered around the coronal midline. Each slice was further partitioned into patches for CNN classification training to detect VCFs, followed by the application of a majority voting mechanism. The method achieved a sensitivity of 88.10% and a specificity of 84.20% on thoracic CT data from 308 patients. However, this study also classified reference standard VCFs based on fractures defined by the Genant classification, without considering acute VCFs. Dong Q. et al. developed a classifier for determining the Genant grade of each vertebral body using the GoogLeNet architecture [34]. This model achieved a sensitivity of 97.7% and a speci-ficity of 95.1% by maximizing Youden’s J statistic in a dataset of 669 patients. However, the acute or chronic status of the fractures was not considered. “

“In alignment with recent research trends in VCF detection, our study analyzed the ef-fectiveness of employing DL techniques for the diagnosis of VCFs. For the EEVD, since ex-tensive preprocessing for efficient feature extraction was not performed, the model identi-fied VCFs directly from patient images, similar to how radiologists approach such cases. While EEVD had the advantage of learning information from the surrounding regions of the vertebral body to detect VCFs, it struggled to focus specifically on radiologic character-istics such as cortical bone information, disruptions, or step-offs that are critical for iden-tifying acute VCFs. Although EEVD demonstrated better performance in identifying VCFs compared to the HLR, it occasionally failed to detect clear VCFs. The TSVD_SD method, in contrast to EEVD, relied on highly limited features extracted from the input images for VCF detection. It exclusively learned features from cortical contour information, making it the most suitable method for identifying VCFs caused by morphological changes. The TSVD_DC method, similar to TSVD_SD, attempted to detect VCFs by limiting the infor-mation used for analysis. This approach involved cropping only the vertebral region for classification, enabling the detection of VCFs based solely on the vertebral image without the influence of information from surrounding regions.

Previous methodologies did not include studies that performed VCF detection using only vertebral contour information as input. This study demonstrated the utility of a deep learning model trained to detect acute VCFs using solely vertebral contour images. It con-firmed the potential of a DL model to detect acute compression fractures even with simple images containing only morphological features of the vertebrae, excluding information from internal vertebral structures, surrounding muscles, or intervertebral discs. The re-sults of this study demonstrated that the TSVD_SD method, which restricted input infor-mation to vertebral contour images, effectively captured the clinical characteristics of acute vertebral fractures. This conclusion was supported by the analysis of the ROC curve, as well as sensitivity and specificity results, which highlighted the clinical relevance of this approach in detecting acute fractures.“

Comments 7: 

The dataset is private and, therefore, the reproducibility of the results raises questions. Authors should provide metrics for an openly available dataset. Additionally, the manuscript fails to include a comparison between this solution and other architectures. Section 4 is, therefore, more appropriate for a review/introduction, as it does not compare the results directly.

Response 7:

Thank you for your constructive comments.

We agreed with the concerns regarding the generalizability of our study.

We added more comments at Limitations part of “4.Discussion” section.

“This study had several limitations. First, this study was conducted using a sin-gle-institution, retrospective dataset, which resulted in a relatively small sample size. This limitation may have introduced selection bias and restricted the generalizability of the findings. The challenge in obtaining a fully labeled large-scale dataset for acute VCFs was primarily due to the time-intensive process of labeling clinical VCF data. Despite this lim-itation, the study provided meaningful insights by investigating clinically classified acute VCFs, even with a small dataset.”

And, as I mentioned above, we added more detailed literature reviews at “4.Discussion” section with Table 4 for comparison between our solution and other architectures.

Reviewer 4 Report

Comments and Suggestions for Authors

Review of “Enhanced detection performance of acute vertebral compression fractures using a hybrid deep learning and traditional quantitative measurement approach: Beyond the limitations of Genant classification” by Jemyoung Lee, Minbeom Kim, Heejun Park, Zepa Yang, Ok Hee Woo, Woo Young Kang and Jong Hyo Kim

The paper examines enhancements in detecting acute vertebral compression fractures (VCFs) using deep learning methods combined with traditional quantitative measurement, challenging the limitations of Genant classification. The authors compared the traditional height loss ratio approach with three deep learning methods for detecting VCFs.

The paper presents intriguing findings. However, a major concern is the opacity of the methods used. The deep learning techniques are described as a “black box”, lacking detailed explanations of the algorithms or coding processes involved. This limitation restricts the ability to fully evaluate the methodology. A more transparent disclosure of the computational methods and codes would greatly enhance the paper’s value.

Additional comments:

1.     The study’s reliance on a retrospective dataset from a single hospital center may introduce selection bias and restrict the generalizability of the results. To enhance the robustness and applicability of the findings, it would be beneficial to discuss parallel studies conducted by other groups working on similar issues. Furthermore, the study's reliance on a single-center retrospective dataset may introduce selection bias and limit generalizability.

2.     The paper delineates criteria for acute VCFs mainly based on imaging characteristics visible in CT scans but lacks detailed descriptions of the initial signs of fracture, such as the initiation of fissures and their specific sizes that qualify for detection by their models.

3.     Another issue raised by the paper is its exclusive focus on the vertebral column without exploring potential correlations with disc degradation. This limitation could affect the applicability of the results in a broader clinical context where interactions between different components of the spinal column are of prime importance. To enhance understanding of spinal health, future research should adopt a multidisciplinary approach that explores the interplay between vertebral and disc health, leading to a more holistic assessment and potentially more effective treatments.

4.     It is suggested to discuss the precise morphological criteria used to classify vertebrae as fractured, including the minimum size of detectable fissures or the presence of other early fracture signs. This detail seems essential for understanding the sensitivity and specificity of the detection models, as minor variations in these criteria can significantly affect the outcome of the diagnosis. Moreover, the inclusion of such specifics would aid in distinguishing acute VCFs from other conditions that might cause similar changes in vertebral morphology, such as Schmorl’s nodes or other degenerative changes, which are often not easily distinguishable in early stages.

Author Response

Thank you for taking the time to review this manuscript with constructive comments.

We have revised the manuscript as you and other reviewers have suggested.

The responses to the comments are as follows.

Comments 1:

The study’s reliance on a retrospective dataset from a single hospital center may introduce selection bias and restrict the generalizability of the results. To enhance the robustness and applicability of the findings, it would be beneficial to discuss parallel studies conducted by other groups working on similar issues. Furthermore, the study's reliance on a single-center retrospective dataset may introduce selection bias and limit generalizability.

Response 1:

Thank you for your constructive comments.

We agreed with the concerns regarding the limitation of retrospective dataset from a single hospital and restriction of generalizability.

We added more comments at Limitations part of “4.Discussion” section.

“This study had several limitations. First, this study was conducted using a single-institution, retrospective dataset, which resulted in a relatively small sample size. This limitation may have introduced selection bias and restricted the generalizability of the findings. The challenge in obtaining a fully labeled large-scale dataset for acute VCFs was primarily due to the time-intensive process of labeling clinical VCF data. Despite this lim-itation, the study provided meaningful insights by investigating clinically classified acute VCFs, even with a small dataset.”

Comments 2: 

The paper delineates criteria for acute VCFs mainly based on imaging characteristics visible in CT scans but lacks detailed descriptions of the initial signs of fracture, such as the initiation of fissures and their specific sizes that qualify for detection by their models.

Response 2:

Thank you for your insightful comments.

We agreed with the concerns regarding the detailed descriptions of the acute VCFs.

To address the reviewer’s comments, we have incorporated additional explanations including a detailed description of VCF at “1.2. Classification of VCF” part of “1.Introduction” section.

“In the acute VCF (typically within the first 6–8 weeks after occurrence), fracture lines were relatively well-defined, and the recent collapse of the vertebral endplates and cortical margins resulted in sharp delineation. At the fracture site, the cortical bone appeared ab-ruptly interrupted or angular, with no evidence of bone bridging [9,10]. The vertebral body often displayed a distinct "step-off" or sharp angulation at the site of the fracture. Due to the absence of bony union, the collapsed regions lacked signs of new bone formation or sclerosis [11,12]. Deformity progression remained possible, and subtle changes in verte-bral alignment occurred with postural adjustments or when mechanical stability was not fully restored.

In chronic VCF, the deformity stabilized and became long-standing. Over time (usu-ally several months), the fracture line gradually became less distinct. The remodeling pro-cess, driven by osteoclastic resorption followed by osteoblastic activity, blurred, smoothed, or even completely obliterated the initial fracture line [13,14]. Mature fractures often ap-peared as stable and well-integrated deformities without a clearly visible fracture line. New bone formation along the endplates and vertebral margins led to sclerotic changes, giving the vertebral body a more uniform and remodeled appearance [13]. The vertebra displayed thickened cortical edges and more regular, rounded contours at the site of the previous fracture. This remodeling reflected a healing response, indicating that the frac-ture was no longer active.

Vertebral fractures were categorized into acute and chronic types. The vertebral height loss calculation method offered the advantage of quantitatively evaluating vertical morphological deformation. However, this approach appeared to have limitations in ef-fectively detecting clinically significant acute fractures, which exhibit more complex de-formities and morphological features, in real-world clinical settings. Furthermore, the re-view of previous studies revealed that research considering acute fractures in the detection of vertebral compression fractures has been underestimated and is relatively rare.”

Comments 3: 

Another issue raised by the paper is its exclusive focus on the vertebral column without exploring potential correlations with disc degradation. This limitation could affect the applicability of the results in a broader clinical context where interactions between different components of the spinal column are of prime importance. To enhance understanding of spinal health, future research should adopt a multidisciplinary approach that explores the interplay between vertebral and disc health, leading to a more holistic assessment and potentially more effective treatments.

Response 3:

We deeply understand and appreciate the reviewer's concerns regarding the limitations of our study.

We agree with the reviewer’s suggestion that it is important to address the potential correlations with disc degeneration rather than focusing exclusively on the vertebrae.

We have added the mentioned points to the limitations in “4.Discussion” section of this study.

“Additionally, while this study focused exclusively on the detection of acute VCFs, ex-ploring potential correlations with disc degeneration is equally important. The applicabil-ity of the study's findings should be considered within the broader clinical context, where interactions among various spinal components play a critical role. To enhance the under-standing of spinal health, future research should adopt a multidisciplinary approach to investigate the interplay between vertebral and disc health. Such an approach would ena-ble a more holistic evaluation and could potentially lead to the development of more effec-tive treatment strategies.”

Comments 4: 

It is suggested to discuss the precise morphological criteria used to classify vertebrae as fractured, including the minimum size of detectable fissures or the presence of other early fracture signs. This detail seems essential for understanding the sensitivity and specificity of the detection models, as minor variations in these criteria can significantly affect the outcome of the diagnosis. Moreover, the inclusion of such specifics would aid in distinguishing acute VCFs from other conditions that might cause similar changes in vertebral morphology, such as Schmorl’s nodes or other degenerative changes, which are often not easily distinguishable in early stages.

Response 4:

We deeply understand and appreciate the reviewer's concerns regarding another limitation of the study.
As suggested by the reviewer, an analysis based on morphological criteria or diseases resembling compression fractures is necessary.
Accordingly, we have incorporated this content into the limitations in “4.Discussion” section of this study.

“specific conditions such as Schmorl’s nodes or other degenerative changes, which can induce morphological alterations similar to vertebral compression fractures (VCFs), were not considered in this study. The exact morphological criteria used to identify vertebrae classified as fractures—particularly the minimum detectable size of fissures or the pres-ence of other early fracture indicators—require further discussion in future studies. Ad-dressing these aspects will be crucial to distinguishing acute VCFs from other conditions with similar morphological features.”

Round 2

Reviewer 1 Report

Comments and Suggestions for Authors

You have made a serious revision; I congratulate you on this.

Reviewer 2 Report

Comments and Suggestions for Authors

Thank you for properly addressing all my comments.The paper is suitable for publication.

Reviewer 3 Report

Comments and Suggestions for Authors

All responses are clear.

Reviewer 4 Report

Comments and Suggestions for Authors

The authors have comprehensively addressed the previous concerns raised during the review process.

It is my opinion that the paper is fully deserving of publication in Bioengineering.